# Robust Transceiver Design for IRS-Assisted Cascaded MIMO Communication Systems

**DOI:** 10.3390/s22176587

**Published:** 2022-08-31

**Authors:** Hossein Esmaeili, Ali Kariminezhad, Aydin Sezgin

**Affiliations:** 1Institute of Digital Communication Systems, Ruhr University Bochum, 44801 Bochum, Germany; 2e:fs TechHub GmbH, 85080 Gaimersheim, Germany

**Keywords:** IRS, robust design, worst case, decode-and-forward, MIMO

## Abstract

Intelligent reconfigurable surfaces (IRSs) have gained much attention due to their passive behavior that can be a successor to relays in many applications. However, traditional relay systems might still be a perfect choice when reliability and throughput are the main concerns in a communication system. In this work, we use an IRS along with a decode-and-forward relay to provide a possible solution to address one of the main challenges of future wireless networks which is providing reliability. We investigate a robust transceiver design against the residual self-interference (RSI), which maximizes the throughput rate under self-interference channel uncertainty-bound constraints. The yielded problem turns out to be a non-convex optimization problem, where the non-convex objective is optimized over the cone of semidefinite matrices. We propose a novel mathematical method to find a lower bound on the performance of the IRS that can be used as a benchmark. Eventually, we show an important result in which, for the worst-case scenario, IRS can be helpful only if the number of IRS elements are at least as large as the size of the interference channel. Moreover, a novel method based on majorization theory and singular value decomposition (SVD) is proposed to find the best response of the transmitters and relay against worst-case RSI. Furthermore, we propose a multi-level water-filling algorithm to obtain a locally optimal solution iteratively. We show that our algorithm performs better that the state of the art in terms of time complexity as well as robustness. For instance, our numerical results show that the acheivable rate can be increased twofold and almost sixfold, respectively, for the case of small and large antenna array at transceivers.

## 1. Introduction

Reliability and throughput are two of the most crucial requirements for the next generation of wireless networks. Optimally relaying the signal from a source to a destination can help enhance reliability and capacity of networks and is currently an active research area [1].

Another emerging candidate for relaying signals is reconfigurable intelligent surfaces (IRSs) [2]. An IRS is a device equipped with multiple passive reconfigurable reflectors that can reflect the colliding waves with an adjustable phase. One of the biggest advantages of IRSs is that they work in a real-time manner without consuming a noticeable amount of power [3]. However, the characteristics of the IRS (e.g., the lack of signal amplification and decode-and-forward processes) can potentially limit its functionality. As a result, in cases where reliability and throughput are of greater importance than power consumption, conventional relays might still be a better option than IRSs. For instance, authors in [4] show that a simple full-duplex relay can outperform an IRS in terms of throughput under certain conditions.

In this paper, we investigate the IRS-assisted MIMO full-duplex (FD) relay system that suffers channel uncertainties. It is also considered that the relays have practical issues such as self-interference (SI) as well as antenna and power limits. The combination of IRS and DF relay can potentially be advantageous. This is due to the fact that both the IRS and the relay have their own limitations that can possibly be compensated for by exploiting each other. The objective of this paper is to maximize the achievable rate of the system by jointly optimizing the impact of the IRS as well as the covariance matrices of the source and the relay.

### 1.1. Related works

IRSs can be utilized in various ways to help the direct links enhance the performance of the system. In [5], authors utilized an IRS to maximize the weighted sum rate MISO system. Authors in [6] proposed a method to minimize the power consumption in a MISO system equipped with an IRS, and authors in [7] investigate the problem of energy efficiency in an IRS-assisted MISO downlink system. The problem of rate maximization in a MIMO system has been presented in [8], where the authors propose an iterative algorithm to find the best IRS pattern assuming the perfect channel state information (CSI) is given. Recently, the authors in [9] utilized IRS in a relay-aided network to minimize the power consumption and successfully showed that the combination of IRS and relay outperforms other scenarios. While the performance of IRS communication systems has been extensively studied, there is not much research that considers robust design when the perfect CSI is not available [10].

Recently, with the emergence of artificial intelligence, IRSs have shown a great potential to improve the existing protocols and technologies [11]. Authors in [12] investigate the benefit of employing an IRS equipped with a multi-task learning system on the transmit power and achievable throughput of aerial–terrestrial communications. In [13], authors use a reinforcement-learning-based approach to optimize the IRS reflection coefficients for buffer-aided relay selection.

One of the earliest studies of the robust transmission designs of IRS-assisted systems was undertaken in [14], where a bounded CSI error model is applied to a problem of power minimization in a MISO transmission system. There, by virtue of semidefinite programming (SDP), the authors turn the original problem into a sequence of convex sub-problems. The robust power minimization subject to the outage probability constraints under statistical cascaded channel error model is considered in [15], where the aim is to optimize the system under worst-case rate constraint. Authors in [16] have proposed a robust algorithm for mean squared error (MSE) minimization for a single user MISO system equipped with an IRS. Their method provides a closed form solution for each iteration. However, it can be used only for the case of a single user system and cannot be extended to more general cases where there are multiple users. Recently, a robust algorithm based on a penalty dual decomposition (PDD) technique is proposed in [10] for sum-rate maximization where they assumed that the channel estimation error follows a complex normal distribution.

Exploiting a relay to improve the communication throughput rate is a classic alternative for IRS in communication systems. However, utilizing a relay in a network raises some important questions to be answered. For instance, how should the relay process the received signal before dispatching it to the destination? Now, the relay can receive a signal from the source, process it and transmit it towards the destination in a successive manner. This type of relaying technique is known as half-duplex relaying. Alternatively, while receiving a signal at a certain time instant, the relay can simultaneously transmit the previously received signals. This technique is known as full-duplex relaying [17].

As a consequence of transmitting and receiving at a common resource unit, the relay is confronted with SI. Note that full-duplex relaying potentially increases the total throughput rate of the communication compared to the half-duplex counterpart only if the SI is handled properly at the relay input. By physically isolating the transmitter and receiver front ends of the relay, a significant portion of SI can be reduced [18]. Moreover, analog and/or digital signal processing at the relay input can be utilized to cancel a portion of SI [19,20,21,22]. This can be realized if the estimate of the SI can be obtained at the relay. These SI cancellation procedures can effectively mitigate the destructive impact of SI up to a certain level. Hence, the remaining portion, the so-called residual self-interference (RSI), is still present at the relay input. The distribution of the RSI is investigated in [23,24]. This RSI is mainly due to the channel estimation uncertainties and transmitter noise. Therefore, the quality of channel estimation plays an important role for limiting RSI if the conventional modulation techniques are utilized.

The authors in [25] employ a superimposed signaling procedure (asymmetric modulation constellation) in the basic point-to-point FD communication for cancelling the SI and further retrieving the desired information contents without requiring channel estimates. They show that for the same average energy per transmission block, the bit error rate of their proposed method is better than that of conventional ones. The RSI evidently degrades the performance of the communication quality. To this end, the authors in [26] study the degrees-of-freedom (DoF), i.e., the slope of the rate curve at asymptotically high SNR and its relation to the performance of an FD cellular network in the presence of RSI. Moreover, the authors in [27,28] investigate the joint rate-energy and delivery time optimization of FD communication, respectively, when RSI is still present. Furthermore, the authors in [29] study the sum rate capacity of the FD channel with and without such degradation. In the presence of RSI, the authors in [30] study the capacity of a Gaussian two-hop FD relay.

Robust transceiver design against the worst-case RSI channel helps find the threshold for switching between HD and FD operating modes. This setup is commonly known as a hybrid relay system [31]. The authors in [32] investigate a robust design for multi-user full-duplex relaying with multi-antenna DF relay. In that work, the sources and destinations are equipped with single antennas. Moreover, the authors in [33] investigate a robust transceiver design for FD multi-user MIMO systems for maximizing the weighted sum-rate of the network. The robust design against worst-case RSI is investigated by authors in [34].

### 1.2. Contribution

Motivated by the above, in this work, we consider a DF multi-hop system with multiple antennas at the source, relay and destination along with an IRS to provide additional links. Then we try to maximize the throughput rate for the worst-case RSI scenario. To the best of our knowledge, this is the first time that the throughput rate maximization against the worst-case RSI is evaluated for IRS-assisted DF full-duplex relay in MIMO systems. First, we simplify the problem by finding an analytical lower bound for the performance of the IRS. Then, the optimization of maximum achievable rate of the DF full-duplex relaying is cast as a non-convex optimization problem. Thereafter, we propose a low complexity method to find the solution using majorization theory. We propose an efficient algorithm to solve this problem in polynomial time. Finally, the transmit signal covariances at the source and the relay are designed efficiently to improve robustness against worst-case RSI channel in a given uncertainty bound. Notice that once the covariances are known, one can easily find the precoders using conventional methods such as singular-value decomposition (SVD), etc. To the best of our knowledge, this is the first work that uses the IRS for RSI cancellation in MIMO full-duplex DF relay systems.

### 1.3. Organization

The rest of the paper is organized as follows. Section 2 outlines the system model and introduces its characteristics. The three different tasks for employing the IRS are also given in this section. In Section 3, the optimization problem belonging to the FD scenario is formulated, and its proper solution is presented. In addition, analytical bounds for the performance of the IRS are given and their corresponding proofs are provided. Section 4 provides the optimization problem for the HD scenario along with the solution. In Section 5, the effectiveness of the proposed algorithm is evaluated and verified by performing numerical simulations over various aspects. Finally, the paper is concluded in Section 6, and technical proofs of the theorems are given in the Appendix A, Appendix B, Appendix C, Appendix D and Appendix E.

## 2. System Model

We consider the communication from a source equipped with $N_{t}$ antennas to a destination with $N_{r}$ antennas. The reliable communication from the transmitter to the destination is assumed to be only feasible by means of a relay with $K_{t}$ transmitter and $K_{r}$ receiver antennas at the output and input front ends, respectively. This means that the direct link from the transmitter to the destination and the link from the transmitter to the IRS and to the destination has a negligible impact on the throughput. This assumption is realistic for the scenarios where the path loss is high due to the high frequency ranges such as mmWave and Terahertz or due to far distances [35], as well as cases were there are objects that block the direct link between the source and the destination. An IRS consisting of *M* elements is established to either cancel the RSI or help enhance one of the transmitter–relay/relay–destination links. The overall system model can be found in Figure 1.

In this paper, it is assumed that signal delivery over the transmitter–IRS–receiver link is not available. This is mainly due to the power attenuation and the power radiation pattern effects [36]. As the IRS is a passive device, it has some power attenuation in practice, which makes the reflected waves weaker than the received ones. In addition, due to the power radiation pattern, based on the angle of arrival and departure, both received and reflected waves are subject to attenuation, respectively. In our system, as the IRS is established in the vicinity of the relay and is faced towards it, both aforementioned effects cause the source–IRS–destination link to be extremely weaker than the source–relay–destination link.

Next, we present the achievable throughput rates for the HD and FD relaying. We start with the HD relay in which κ=0. In the second case, IRS can be exploited to enhance the quality of the channel between the source and the relay. In such a case, the received signals at the relay and destination can be expressed as
(1)$$\begin{array}{cc} y_{r} & =\left(H_{1}+H_{IR}\Theta H_{SI}\right)x_{s}+\kappa H_{r}x_{r}+n_{t}, \end{array}$$
(2)$$\begin{array}{cc} y_{d} & =H_{2}x_{r}+n_{d}, \end{array}$$
where $H_{SI}\in C^{N_{t}M}$ is the channel from the source to the IRS. Finally, IRS can be used to help the channel from the relay to the destination. In this case, the received signals are going to be
(3)$$\begin{array}{cc} y_{r} & =H_{1}x_{s}+\kappa H_{r}x_{r}+n_{t}, \end{array}$$
(4)$$\begin{array}{cc} y_{d} & =\left(H_{2}+H_{ID}\Theta H_{RI}\right)x_{r}+n_{d}, \end{array}$$
where $H_{ID}\in C^{M\times N_{r}}$ is the channel from the source to the IRS.

In what follows, we find the achievable rate for three aforementioned cases and compare them to see under what conditions each of them should be applied. Notation and definitions are summarized in Table 1.

## 3. Achievable Rate (Full-Duplex Relay)

### 3.1. Overview

Suppose that the relay employs a DF strategy. In the full-duplex scenario, both source–relay and relay–destination links are active at the same time. As a result, the signals from the relay transmitter interfere with the receiving signal at the relay receiver. We assume that an estimate of the SI channel $H_{r}$ is available at the relay denoted by ${\hat{H}}_{r}$. Hence, the RSI represented by ${\bar{H}}_{r}$ is given as
(5)$$\begin{array}{c} {\bar{H}}_{r}=H_{r}-{\hat{H}}_{r}. \end{array}$$
In the rest of the paper, we try to find an approach to deal with this RSI.

### 3.2. Mathematical Preliminaries

Considering a FD DF relay, the following rates are achievable [37],
(6)$$\begin{array}{c} R^{\mathrm{FD}}=\min\left(R_{\mathrm{sr}}^{\mathrm{FD}},R_{\mathrm{rd}}^{\mathrm{FD}}\right), \end{array}$$
in which, depending on how the IRS is applied to the system, the three following sets of rates are possible. First,
(7)$$\begin{array}{cc} R_{\mathrm{sr}}^{\mathrm{FD}} & =\log_{2}\frac{\left|\sigma_{t}^{2}I_{K_{r}}+{\hat{H}}_{1}Q_{s}{\hat{H}}_{1}^{H}+{\bar{H}}_{1}Q_{s}{\bar{H}}_{1}^{H}+H_{tot_{1}}Q_{r}H_{tot_{1}}^{H}\right|}{\left|\sigma_{t}^{2}I_{K_{r}}+{\bar{H}}_{1}Q_{s}{\bar{H}}_{1}^{H}+H_{tot_{1}}Q_{r}H_{tot_{1}}^{H}\right|}, \end{array}$$
(8)$$\begin{array}{cc} R_{\mathrm{rd}}^{\mathrm{FD}} & =\log_{2}\frac{\left|\sigma_{d}^{2}I_{N}+{\hat{H}}_{2}Q_{r}{\hat{H}}_{2}^{H}+{\bar{H}}_{2}Q_{r}{\bar{H}}_{2}^{H}\right|}{\left|\sigma_{d}^{2}I_{N}+{\bar{H}}_{2}Q_{r}{\bar{H}}_{2}^{H}\right|}, \end{array}$$
where $H_{tot_{1}}=\left({\bar{H}}_{r}+H_{RI}\Theta H_{IR}\right)$ when the IRS is used to cancel the self interference. Second,
(9)$$\begin{array}{cc} R_{\mathrm{sr}}^{\mathrm{FD}} & =\log_{2}\frac{\left|\sigma_{t}^{2}I_{K_{r}}+{\hat{H}}_{tot_{2}}Q_{s}{\hat{H}}_{tot_{2}}^{H}\;+\;{\bar{H}}_{tot_{2}}Q_{s}{\bar{H}}_{tot_{2}}^{H}+{\bar{H}}_{r}Q_{r}{\bar{H}}_{r}^{H}\right|}{\left|\sigma_{t}^{2}I_{K_{r}}\;+\;{\bar{H}}_{tot_{2}}Q_{s}{\bar{H}}_{tot_{2}}^{H}+{\bar{H}}_{r}Q_{r}{\bar{H}}_{r}^{H}\right|}, \end{array}$$
(10)$$\begin{array}{cc} R_{\mathrm{rd}}^{\mathrm{FD}} & =\log_{2}\frac{\left|\sigma_{d}^{2}I_{N}+{\hat{H}}_{2}Q_{r}{\hat{H}}_{2}^{H}+{\bar{H}}_{2}Q_{r}{\bar{H}}_{2}^{H}\right|}{\left|\sigma_{d}^{2}I_{N}+{\bar{H}}_{2}Q_{r}{\bar{H}}_{2}^{H}\right|}, \end{array}$$
where ${\hat{H}}_{tot_{2}}=\left({\hat{H}}_{1}+{\hat{H}}_{RI}\Theta {\hat{H}}_{IR}\right)$ and ${\bar{H}}_{tot_{2}}=\left({\bar{H}}_{1}+{\bar{H}}_{RI}\Theta {\bar{H}}_{IR}\right)$ if the IRS is established to help the source–relay channel and finally
(11)$$\begin{array}{cc} R_{\mathrm{sr}}^{\mathrm{FD}} & =\log_{2}\frac{\left|\sigma_{t}^{2}I_{K_{r}}+{\hat{H}}_{1}Q_{s}{\hat{H}}_{1}^{H}+{\bar{H}}_{1}Q_{s}{\bar{H}}_{1}^{H}+{\bar{H}}_{r}Q_{r}{\bar{H}}_{r}^{H}\right|}{\left|\sigma_{t}^{2}I_{K_{r}}+{\bar{H}}_{1}Q_{s}{\bar{H}}_{1}^{H}+{\bar{H}}_{r}Q_{r}{\bar{H}}_{r}^{H}\right|}, \end{array}$$
(12)$$\begin{array}{cc} R_{\mathrm{rd}}^{\mathrm{FD}} & =\log_{2}\frac{\left|\sigma_{d}^{2}I_{N}+{\hat{H}}_{tot_{3}}Q_{r}{\hat{H}}_{tot_{3}}^{H}+{\bar{H}}_{tot_{3}}Q_{r}{\bar{H}}_{tot_{3}}^{H}\right|}{\left|\sigma_{d}^{2}I_{N}+{\bar{H}}_{tot_{3}}Q_{r}{\bar{H}}_{tot_{3}}^{H}\right|}, \end{array}$$
where ${\hat{H}}_{tot_{3}}=\left({\hat{H}}_{2}+{\hat{H}}_{RI}\Theta {\hat{H}}_{IR}\right)$ and ${\bar{H}}_{tot_{3}}=\left({\bar{H}}_{2}+{\bar{H}}_{RI}\Theta {\bar{H}}_{IR}\right)$ if the IRS is utilized to enhance the rate of the relay–destination channel. Notice that assuming that the RSI remains uncanceled, a robust transceiver against the worst-case RSI channel is required which is formulated as an optimization problem as follows
(13)$$\begin{array}{cc} \max_{Q_{s},Q_{r},\Theta }\;\min_{{\bar{H}}_{r}}\; & \min\left(R_{\mathrm{sr}}^{\mathrm{FD}},R_{\mathrm{rd}}^{\mathrm{FD}}\right) \end{array}$$
(13a)$$\begin{array}{cc} \mathrm{subject}\;\mathrm{to}\;\; & \mathrm{Tr}\left(Q_{s}\right)\leq P_{s}, \end{array}$$
(13b)$$\begin{array}{cc} \; & \mathrm{Tr}\left(Q_{r}\right)\leq P_{r},\\ \; & \mathrm{Tr}\left({\bar{H}}_{x}{\bar{H}}_{x}^{H}\right)\leq T_{x}, \end{array}$$
(13c)$$\begin{array}{cc} \; & x\in \left\{1,2,r,RI,IR,ID,SI\right\} \end{array}$$
(13d)$$\begin{array}{cc} \; & |\theta_{m}|\leq 1,\forall m \end{array}$$
in which the throughput rate with respect to the worst-case RSI channel is maximized. Two constraints, $P_{s}$ and $P_{r}$, represent the transmit power budgets at the source and the relay, respectively. In constraint (13c), $T_{x}$ represents the RSI or the channel estimation error bound corresponding to $H_{x}$. Notice that $\mathrm{Tr}({\bar{H}}_{x}{\bar{H}}_{x}^{H})$ represents the sum of the squared singular values of $H_{x}$. It should be noted that using a bounded matrix norm is the most common way for modeling the uncertainty of a matrix [38,39]. In practice, $T_{x}$ can be found using stochastic methods when the distribution of the channel error is known. Otherwise, one may find it using a sample average approximation method. Finally, constraints (13d) are due to the unit modulus limitation of the IRS elements.

The problem (Equation 13) is non-convex and hard to solve. As a result, for each of the above-mentioned scenarios, we propose a simplified version of the optimization problem and try to solve it instead. Note that as we are interested in finding the throughput corresponding to the worst-case RSI, any simplification in the optimization problem should be in favor of the RSI and interference. First, we analyse the performance of the system when the IRS is helping the relay cancel the RSI. Consequently, we show that the problem (Equation 13) can be simplified to the following optimization problem
(14)$$\begin{array}{cc} \max_{Q_{s},Q_{r}}\;\min_{H_{tot}}\; & \min\left(R_{\mathrm{sr}}^{\mathrm{FD}},R_{\mathrm{rd}}^{\mathrm{FD}}\right) \end{array}$$
(14a)$$\begin{array}{cc} \mathrm{subject}\;\mathrm{to}\;\; & \mathrm{Tr}\left(Q_{s}\right)\leq P_{s},\\ \; & \mathrm{Tr}\left(Q_{r}\right)\leq P_{r},\\ \; & \mathrm{Tr}\left(H_{tot}H_{tot}^{H}\right)\leq T^{\prime }\left(T_{r},\Theta \right).\\ \; & \mathrm{Tr}\left({\bar{H}}_{x}{\bar{H}}_{x}^{H}\right)\leq T_{x},\;x\in \left\{1,2\right\} \end{array}$$
where
(15)$$\begin{array}{cc} T^{\prime }\left(T_{r},\Theta \right)=\min_{\Theta }\;\max_{{\bar{H}}_{r}}\; & \left|\right|{\bar{H}}_{r}+H_{RI}\Theta H_{IR}{\left|\right|}_{F}^{2} \end{array}$$
(15a)$$\begin{array}{cc} \mathrm{subject}\;\mathrm{to}\;\;\; & \mathrm{Tr}\left({\bar{H}}_{r}{\bar{H}}_{r}^{H}\right)\leq T_{r}, \end{array}$$
(15b)$$\begin{array}{cc} \; & {\left||Vec\left(\Theta \right)|\right|}_{2}^{2}\leq 1, \end{array}$$
and where Vec(·) denotes the vector of all non-zero elements of its input matrix. We can equivalently write $T^{\prime }$ as
(16)$$\begin{array}{cc} T^{\prime }\left(T_{r},\Theta \right)=\min_{\Theta }\;\max_{{\bar{H}}_{r}}\; & ||Vec\left({\bar{H}}_{r}\right)+\left(H_{IR}*H_{RI}^{T}\right)Vec\left(\Theta \right){\left|\right|}_{2}^{2} \end{array}$$
(16a)$$\begin{array}{cc} \mathrm{subject}\;\mathrm{to}\;\;\; & \mathrm{Tr}\left({\bar{H}}_{r}{\bar{H}}_{r}^{H}\right)\leq T_{r}, \end{array}$$
(16b)$$\begin{array}{cc} \; & {\left||Vec\left(\Theta \right)|\right|}_{2}^{2}\leq 1, \end{array}$$
where ∗ denotes a column-wise Khatri–Rao product defined as below
(17)$$A*B=\left[A_{1}⊗B_{1}|A_{2}⊗B_{2}\left|\cdots \right|A_{n}⊗B_{n}\right],$$
and where $A_{i}$ is the *i*’th column of *A*, and ⊗ denotes the Kronecker product. See Appendix A for proof.

One can show that $T^{\prime }\leq {\left(\sqrt{T_{r}}-\sigma_{\min}\left(H_{IR}*H_{RI}^{T}\right)\right)}^{2}$. As mentioned before, problem (14) is a simplification of problem (Equation 13). This means every achievable rate which is inside the feasible set of (14) is also inside the feasible set of (Equation 13) (Notice that the reverse is not necessarily true, i.e., every achievable rate which is a feasible solution of (Equation 13) is not necessarily a feasible solution for (14) as well. However, as we look for achievable rates, we can still use this method). The reason is that in problem (Equation 13), the minimization over RSI happens only one time, and the worst-case RSI simultaneously tries to cancel the effect of the best configuration of IRS and the best covariance matrices. In (14), first, the RSI does its worst damage on the performance of the best IRS configuration and after that performs another optimization to bring the worst power allocation against the best covariance matrices (This will be clearer later on when the geometrical representation of the problem is given). In what follows, we provide our proposed ways to deal with optimization problems (14) and (15), respectively.

**Theorem** **1.**
*For the optimization problem (14), one can show that $T^{\prime }\left(T,\Theta \right)\leq {\left(\sqrt{T_{r}}-\sigma_{\min}\left(H_{IR}*H_{RI}^{T}\right)\right)}^{2}$.*


**Proof.** We begin the proof with an intuitive example and then extend it to the more general case. Assume that $K_{t}=1,K_{r}=2$ and M=3. Then we have
(18)$$\begin{array}{cc} T^{\prime }=\max_{\Theta }\;\min_{{\bar{h}}_{r}}\; & \left|\right|{\bar{h}}_{r}+H_{IR}diag\left(H_{RI}^{T}\right)Vec\left(\Theta \right){\left|\right|}_{2}^{2} \end{array}$$
(18a)$$\begin{array}{cc} \mathrm{subject}\;\mathrm{to}\;\;\; & {\bar{h}}_{11}^{2}+{\bar{h}}_{21}^{2}\leq T_{r}, \end{array}$$
(18b)$$\begin{array}{cc} \; & \theta_{1}^{2}\leq 1,\;\theta_{2}^{2}\leq 1,\;\theta_{3}^{2}\leq 1. \end{array}$$In addition, consider the following optimization problem
(19)$$\begin{array}{cc} T^{\prime \prime }=\max_{\Theta }\;\min_{{\bar{h}}_{r}}\; & \left|\right|{\bar{h}}_{r}+H_{IR}diag\left(H_{RI}^{T}\right)Vec\left(\Theta \right){\left|\right|}_{2}^{2} \end{array}$$
(19a)$$\begin{array}{cc} \mathrm{subject}\;\mathrm{to}\;\;\; & {\bar{h}}_{11}^{2}+{\bar{h}}_{21}^{2}\leq T_{r}, \end{array}$$
(19b)$$\begin{array}{cc} \; & \theta_{1}^{2}+\theta_{2}^{2}+\theta_{3}^{2}\leq 1, \end{array}$$Here, notice that $H_{IR}diag\left(H_{RI}^{T}\right)$ is a linear map from a three-dimensional into a two-dimensional space. One simple example of such a mapping can be found in Figure 2. Here, an example of mapping from a three-dimensional to a two-dimensional space is shown. The left shape shows the feasible set for the IRS with three elements in a real valued space. The cube belongs to the case of $T^{\prime }$, i.e., constraints $-1\leq \theta_{m}\leq 1,\forall m$, while the sphere shows the constraint $\theta_{1}^{2}+\theta_{2}^{2}+\theta_{3}^{2}\leq 1$ which belongs to $T^{\prime \prime }$. On the right, the feasible sets belonging to the two aforementioned regions after performing mapping *f* are presented as an example. It can be seen that the mapping of the first set of constraints (the hexagon) covers the whole area of that of the second one (the ellipse). One important key is, as mapping is a linear function, we have $A\subset B\overset{}{\to }f\left(A\right)\subset f\left(B\right)$, where A and B are two arbitrary sets and *f* is the mapping.In general, as the number of IRS elements or the dimensions of ${\bar{h}}_{r}$ increase, the mapping of the hypercube becomes more and more complicated and finding the optimal distance becomes more difficult. However, there is an upper bound for this distance. As shown in Figure 3, if instead of the cube, we limit the feasible set of IRS elements to the sphere inside the cube, i.e., replacing (18b) with (19b), the solution to the problem becomes GE≥GF. It turns out that finding GE is very simple as by the definition we have $\sigma_{\min}\left(H_{IR}diag\left(H_{RI}^{T}\right)\right)=OE$, and also we know that $\sqrt{T_{r}}=GO$. Therefore, we can conclude $GE=\sqrt{T_{r}}-\sigma_{\min}\left(H_{IR}diag\left(H_{RI}^{T}\right)\right)$. Finally, we use one last upper bound to make the original problem even easier to solve. Note that if instead of the ellipse, we consider the circle inscribed in it, we will have $\max_{{\bar{h}}_{r}}\;\min_{\Theta }\left|\right|{\bar{h}}_{r}+\left(H_{IR}*H_{RI}^{T}\right)\theta {\left|\right|}_{2}=\sqrt{T_{r}}-\sigma_{\min}\left(H_{IR}*H_{RI}^{T}\right),\forall {\bar{h}}_{r}$. As a result, we have
(20)$$\left|\right|{\bar{h}}_{tot}{\bar{h}}_{tot}^{H}{\left|\right|}_{2}\leq T^{\prime },\;\forall {\bar{h}}_{r},$$
where $T^{\prime }={\left(\sqrt{T_{r}}-\sigma_{\min}\left(H_{IR}*H_{RI}^{T}\right)\right)}^{2}$. It is worth mentioning that the geometrical representation for the optimization problem (Equation 13) is different because there, considering that the RSI wants to bring the worst representation against the IRS configuration and covariance matrices simultaneously, the RSI cannot freely span the whole circle. This is due to the fact that some regions in the circle might not be a good choice when it comes to RSI design against covariance matrices. However, if the best representation of RSI against the covariance matrices also provides the best RSI against the IRS configuration, the solution to (14) and (Equation 13) will be the same. Eventually, instead of optimization problem (Equation 13), one can solve optimization problem (14). The solution to the new problem is guaranteed to be achievable by the original problem as well.Notice that one can readily extend this interpretation into the complex domain, as the constraint (19b) will still be a subset of constraints (18b). It remains to show one can generalize the geometrical proof for arbitrary large dimensions. This means that the channel dimensions and the number of IRS elements can be any natural numbers. Interestingly, it is enough to show that the geometrical proof based on $\ell_{2}$ norms and Euclidean distance exists for higher dimensions. This proof is given in Appendix A where it is shown $\left|\right|{\bar{H}}_{r}+H_{RI}\Theta H_{IR}{\left|\right|}_{F}^{2}=||Vec\left({\bar{H}}_{r}\right)+\left(H_{IR}*H_{RI}^{T}\right)Vec\left(\Theta \right){\left|\right|}_{2}^{2}$.    □

Next we solve problem (14). Solving this problem is hard in general as it is non-convex. Hence, we first use the following lemma and theorem to solve it. There, it is shown that for every possible choice of $H_{1}$ and $H_{2}$, there exists at least one set of simultaneously diagonalizable matrices $H_{tot}$, $Q_{s}$ and $Q_{r}$ that are the solutions to the problem (14).

**Lemma** **1.**
*For two positive semi-definite and positive definite matrices A and B with eigenvalues $\lambda_{1}\left(A\right)\geq \lambda_{2}\left(A\right)\geq ...\geq \lambda_{N}\left(A\right)$ and $\lambda_{1}\left(B\right)\geq \lambda_{2}\left(B\right)\geq ...\geq \lambda_{N}\left(B\right)$, respectively, the following inequalities hold,*

(21)
$$\prod_{i=1}^{N}\left(1+\frac{\lambda_{i}\left(A\right)}{\lambda_{i}\left(B\right)}\right)\leq |I+AB^{-1}|\leq \prod_{i=1}^{N}\left(1+\frac{\lambda_{i}\left(A\right)}{\lambda_{N+1-i}\left(B\right)}\right).$$



**Proof.** Consider Fiedler’s inequality given by [40],
(22)$$\prod_{i=1}^{N}\left(\lambda_{i}\left(A\right)\;+\;\lambda_{i}\left(B\right)\right)\;\leq \;|B+A|\;\leq \;\prod_{i=1}^{N}\left(\lambda_{i}\left(A\right)\;+\;\lambda_{N+1-i}\left(B\right)\right).$$Furthermore, given B as a positive definite matrix, the following are true,
(23)$$\begin{array}{cc} \left|B^{}\right| & >0, \end{array}$$
(24)$$\begin{array}{cc} \left|B^{-1}\right| & =\prod_{i=1}^{N}\frac{1}{\lambda_{i}\left(B\right)}. \end{array}$$
Now, dividing the sides of (22) by |B|, one can readily obtain (21).    □

Note that in (21), the inequalities hold with equality if and only if A and B are diagonalizable over a common basis. Using the result of Lemma 1, $R_{\mathrm{sr}}^{\mathrm{FD}}$ can be lower-bounded as
(25)$$\begin{array}{cc} \; & \log_{2}\frac{\left|\sigma_{t}^{2}I_{K_{r}}+{\hat{H}}_{1}Q_{s}{\hat{H}}_{1}^{H}+{\bar{H}}_{1}Q_{s}{\bar{H}}_{1}^{H}+H_{tot_{1}}Q_{r}H_{tot_{1}}^{H}\right|}{\left|\sigma_{t}^{2}I_{K_{r}}+{\bar{H}}_{1}Q_{s}{\bar{H}}_{1}^{H}+H_{tot_{1}}Q_{r}H_{tot_{1}}^{H}\right|}\geq \\ \; & \sum_{i=1}^{\min\left(M,K_{r}\right)}\;\log_{2}\;\left(\;1+\;\frac{\lambda_{i}\left(H_{1}Q_{s}{H_{1}}^{H}\right)}{\lambda_{i}\;\left(\;\sigma_{t}^{2}I_{K_{r}}\;+\;{\bar{H}}_{1}\;Q_{s}\;{\bar{H}}_{1}^{H}\;+\;H_{tot_{1}}\;Q_{r}\;H_{tot_{1}}^{H}\;\right)}\right). \end{array}$$

In addition, it holds that $\lambda_{i}\;\left(\;\sigma_{t}^{2}I_{K_{r}}\;+\;{\bar{H}}_{1}\;Q_{s}\;{\bar{H}}_{1}^{H}\;+\;H_{tot_{1}}\;Q_{r}\;H_{tot_{1}}^{H}\;\right)=\;\sigma_{t}^{2}\;+$$\lambda_{i}\;\left(\;{\bar{H}}_{1}\;Q_{s}\;{\bar{H}}_{1}^{H}\;+\;H_{tot_{1}}\;Q_{r}\;H_{tot_{1}}^{H}\;\right)$. Hence, we obtain
(26)$$\begin{array}{cc} \; & \log_{2}\frac{\left|\sigma_{t}^{2}I_{K_{r}}+{\hat{H}}_{1}Q_{s}{\hat{H}}_{1}^{H}+{\bar{H}}_{1}Q_{s}{\bar{H}}_{1}^{H}+H_{tot_{1}}Q_{r}H_{tot_{1}}^{H}\right|}{\left|\sigma_{t}^{2}I_{K_{r}}+{\bar{H}}_{1}Q_{s}{\bar{H}}_{1}^{H}+H_{tot_{1}}Q_{r}H_{tot_{1}}^{H}\right|}\geq \\ \; & \sum_{i=1}^{\min\left(M,K_{r}\right)}\;\log_{2}\;\left(\;1+\;\frac{\lambda_{i}\left(H_{1}Q_{s}{H_{1}}^{H}\right)}{\;\sigma_{t}^{2}\;+\;\lambda_{i}\;\left(\;{\bar{H}}_{1}\;Q_{s}\;{\bar{H}}_{1}^{H}\;+\;H_{tot_{1}}\;Q_{r}\;H_{tot_{1}}^{H}\;\right)}\right). \end{array}$$

Note that the inequality holds with equality whenever $H_{1}Q_{s}{H_{1}}^{H}$ and ${\bar{H}}_{1}\;Q_{s}\;{\bar{H}}_{1}^{H}\;+\;H_{tot_{1}}\;Q_{r}\;H_{tot_{1}}^{H}$ share a common basis. Next, we use the following inequality
(27)$$\begin{array}{cc} \; & \sum_{i=1}^{\min\left(M,K_{r}\right)}\;\log_{2}\;\left(\;1+\;\frac{\lambda_{i}\left(H_{1}Q_{s}{H_{1}}^{H}\right)}{\;\sigma_{t}^{2}\;+\;\lambda_{i}\;\left(\;{\bar{H}}_{1}\;Q_{s}\;{\bar{H}}_{1}^{H}\;+\;H_{tot_{1}}\;Q_{r}\;H_{tot_{1}}^{H}\;\right)}\right)\geq \\ \; & \sum_{i=1}^{\min\left(M,K_{r}\right)}\;\log_{2}\;\left(\;1+\;\frac{\lambda_{i}\left(H_{1}Q_{s}{H_{1}}^{H}\right)}{\;\sigma_{t}^{2}\;+T_{1}P_{s}+\lambda_{i}\;\left(\;H_{tot_{1}}\;Q_{r}\;H_{tot_{1}}^{H}\;\right)}\right). \end{array}$$

The above inequality holds true since $\lambda_{i}\left({\bar{H}}_{1}Q_{s}{\bar{H}}_{1}^{H}\right)\leq T_{1}P_{s}$.

Now, instead of completing the minimization over the left-hand side (LHS) of Equation (26), we can first minimize the right-hand side (RHS) of Equation (27) to find an achievable rate. Similarly, for $R_{\mathrm{rd}}^{\mathrm{FD}}$ we have
(28)$$\begin{array}{cc} \; & \log_{2}\frac{\left|\sigma_{d}^{2}I_{N}+{\hat{H}}_{2}Q_{r}{\hat{H}}_{2}^{H}+{\bar{H}}_{2}Q_{r}{\bar{H}}_{2}^{H}\right|}{\left|\sigma_{d}^{2}I_{N}+{\bar{H}}_{2}Q_{r}{\bar{H}}_{2}^{H}\right|}\geq \sum_{i=1}^{\min\left(M,K_{r}\right)}\;\log_{2}\;\left(\;1+\;\frac{\lambda_{i}\left(H_{2}Q_{r}{H_{2}}^{H}\right)}{\;\sigma_{d}^{2}+T_{2}P_{r}}\right). \end{array}$$

**Remark** **1.**
*Having the equality $C={\bar{H}}_{r}Q_{r}{\bar{H}}_{r}^{H}$, one can generally conclude the rule of multiplication is determinant, i.e., $\det\left(C\right)=\det\left({\bar{H}}_{r}^{H}{\bar{H}}_{r}\right)\det\left(Q_{r}\right)$. Further, using the properties of determinants we can also conclude $\prod_{i=1}^{N}\lambda_{i}\left(C\right)=\prod_{i=1}^{N}\left(\lambda_{\rho \left(i\right)}\left({\bar{H}}_{r}^{H}{\bar{H}}_{r}\right)\lambda_{i}\left(Q_{r}\right)\right)$ where $\rho \left(i\right)$ is a random permutation of i and indicates that there is no need for $\lambda_{\rho \left(i\right)}\left({\bar{H}}_{r}^{H}{\bar{H}}_{r}\right)$ to be in decreasing order. However, one cannot generally conclude $\lambda_{i}\left(C\right)=\lambda_{\rho \left(i\right)}\left({\bar{H}}_{r}^{H}{\bar{H}}_{r}\right)\lambda_{i}\left(Q_{r}\right),\forall i$, unless ${\bar{H}}_{r}^{H}{\bar{H}}_{r}$ and $Q_{r}$ share common basis.*


As a result of Remark 1, in general, we cannot rewrite (26) in terms of $\lambda_{i}\left({\bar{H}}_{r}^{H}{\bar{H}}_{r}\right)$, $\lambda_{i}\left(Q_{r}\right)$, $\lambda_{i}\left(H_{1}^{H}H_{1}\right)$ and $\lambda_{i}\left(Q_{s}\right)$. However, if we show that for every choice of $Q_{s}$, there exists a matrix $Q_{s}^{\prime }$ with properties: (1) $\lambda_{i}\left(H_{1}Q_{s}H_{1}^{H}\right)=\lambda_{i}\left(H_{1}Q_{s}^{\prime }H_{1}^{H}\right)$; (2) $\lambda_{i}\left(H_{1}Q_{s}^{\prime }H_{1}^{H}\right)=\lambda_{i}\left(Q_{s}^{\prime }\right)\lambda_{i}\left(H_{1}^{H}H_{1}\right)$ and 3) $\mathrm{Tr}\left(Q_{s}^{\prime }\right)\leq \mathrm{Tr}\left(Q_{s}\right)$; then we can use $Q_{s}^{\prime }$ instead and rewrite  (26) in terms of $\lambda_{i}\left(H_{1}^{H}H_{1}\right)$ and $\lambda_{i}\left(Q_{s}^{\prime }\right)$ to simplify the problem. The first property implies that both $Q_{s}$ and $Q_{s}^{\prime }$ have the exact same impact on the capacity. Hence, if we find a $Q_{s}$ which is the solution to the problem (14), its corresponding $Q_{s}^{\prime }$ will also be a solution. The second property means, unlike $Q_{s}$, $Q_{s}^{\prime }$ actually shares the common basis with $H_{1}^{H}H_{1}$. The last property implies that $Q_{s}^{\prime }$ is at least as good as $Q_{s}$ in terms of power consumption. Observe that if we show for every feasible $Q_{s}$ there exists at least one such $Q_{s}^{\prime }$, then we can solve the problem (14) in a much easier way. The reason is, in such a case, instead of searching for optimal $Q_{s}$ over the whole feasible set, we can search for the optimal $Q_{s}^{\prime }$. Unlike $Q_{s}$, finding $Q_{s}^{\prime }$ does not need a complete search over the whole feasible set since $Q_{s}^{\prime }$ shares a common basis with $H_{1}^{H}H_{1}$. Therefore, we can limit our search only to the portion of the feasible set in which the matrices have eigendirections identical to those of $H_{1}^{H}H_{1}$. Similarly, if we show for every choice of ${\bar{H}}_{r}$, there exist at least one ${\bar{H}}_{r}^{\prime }$ for which we have three conditions $\lambda_{i}\left({\bar{H}}_{r}Q_{r}{\bar{H}}_{r}^{H}\right)=\lambda_{i}\left({\bar{H}}_{r}^{\prime }Q_{r}{\bar{H}}_{r}^{\prime H}\right)$, $\lambda_{i}\left({\bar{H}}_{r}Q_{r}{\bar{H}}_{r}^{\prime H}\right)=\lambda_{i}\left(Q_{r}\right)\lambda_{i}\left({\bar{H}}_{1}^{\prime H}{\bar{H}}_{1}^{\prime }\right)$ and $\mathrm{Tr}\left({\bar{H}}_{1}^{{}^{\prime }H}{\bar{H}}_{1}^{\prime }\right)\leq \mathrm{Tr}\left({\bar{H}}_{1}^{H}{\bar{H}}_{1}\right)$, we can simplify our search to finding ${\bar{H}}_{r}^{\prime }$ instead of ${\bar{H}}_{r}$. In the next theorem, we show that such $Q_{s}^{\prime }$ and ${\bar{H}}_{r}^{\prime }$ exist.

**Theorem** **2.**
*For all matrices $Q_{s}$ and $H_{1}$, there exists at least one matrix $Q_{s}^{\prime }$ that satisfies the following conditions,*

(29)
$$\begin{array}{cc} \lambda_{i}\left(H_{1}Q_{s}H_{1}^{H}\right) & =\lambda_{i}\left(H_{1}Q_{s}^{\prime }H_{1}^{H}\right), \end{array}$$


(30)
$$\begin{array}{cc} \lambda_{i}\left(H_{1}Q_{s}^{\prime }H_{1}^{H}\right) & =\lambda_{\rho \left(i\right)}\left(Q_{s}^{\prime }\right)\lambda_{i}\left(H_{1}^{H}H_{1}\right), \end{array}$$


(31)
$$\begin{array}{cc} \mathrm{Tr}(Q_{s}^{\prime }) & \leq \mathrm{Tr}(Q_{s}), \end{array}$$

*where $\rho \left(i\right)$ is a random permutation of i and indicates that there is no need for $\lambda_{\rho \left(i\right)}\left(Q_{s}^{\prime }\right)$ to be in decreasing order.*


**Proof.** The proof is given in Appendix B.    □

For the sake of simplicity, we use the following notions for the rest of the paper,
(32)$$\begin{array}{cc} \; & \gamma_{s_{i}}=\lambda_{i}\left(Q_{s}\right), \end{array}$$
(33)$$\begin{array}{cc} \; & \gamma_{r_{i}}=\lambda_{i}\left(Q_{r}\right), \end{array}$$
(34)$$\begin{array}{cc} \; & \sigma_{1_{i}}^{2}=\lambda_{i}\left(H_{1}H_{1}^{H}\right), \end{array}$$
(35)$$\begin{array}{cc} \; & \sigma_{r_{i}}^{2}=\lambda_{i}\left({\bar{H}}_{tot_{1}}{\bar{H}}_{tot_{1}}^{H}\right), \end{array}$$
(36)$$\begin{array}{cc} \; & \sigma_{2_{i}}^{2}=\lambda_{i}\left(H_{2}H_{2}^{H}\right). \end{array}$$

Now, using Theorem 2 alongside Lemma 1, we infer that with no loss of generality, instead of optimising over matrices, one can complete the optimization over eigenvalues to find the optimal value for RSH of (27). Then we have
$$\begin{array}{cc} \max_{\gamma_{s},\gamma_{r}}\;\min_{\sigma_{r}}\; & \min\;(\;\sum_{i=1}^{\min(M,K_{r})}\;\log\;\left(\;1+\;\frac{\sigma_{1_{i}}^{2}\gamma_{s_{\rho \left(i\right)}}}{\sigma_{t}^{2}\;+T_{1}P_{s}+\gamma_{r_{i}}\sigma_{r_{\rho \left(i\right)}}^{2}}\right), \end{array}$$
(37)$$\begin{array}{c} \;\sum_{i=1}^{\min(K_{t},N)}\;\log\;\left(1+\frac{\sigma_{2_{i}}^{2}\gamma_{r_{i}}}{\sigma_{d}^{2}+T_{2}P_{r}}\right)) \end{array}$$
(37a)$$\begin{array}{cc} \mathrm{subject}\;\mathrm{to}\; & ∥\gamma_{s}{∥}_{1}\leq P_{s}, \end{array}$$
(37b)$$\begin{array}{cc} \; & ∥\gamma_{r}{∥}_{1}\leq P_{r}, \end{array}$$
(37c)$$\begin{array}{cc} \; & ∥\sigma_{r}^{2}{∥}_{1}\leq T^{\prime }, \end{array}$$
(37d)$$\begin{array}{cc} \; & \sigma_{1_{i}}^{2}\gamma_{s_{\rho \left(i\right)}}\geq \sigma_{1_{i+1}}^{2}\gamma_{s_{\rho \left(i+1\right)}},\;\forall i\leq \min\left(M,K_{r}\right), \end{array}$$
(37e)$$\begin{array}{cc} \; & \gamma_{r_{i}}\sigma_{r_{\rho \left(i\right)}}^{2}\geq \gamma_{r_{i+1}}\sigma_{r_{\rho \left(i+1\right)}}^{2},\;\forall i\leq \min(K_{t},N). \end{array}$$
Note that the two additional constraints (37d) and (37e) need to be satisfied due to the conditions of Lemma 1 (i.e., eigenvalues have to be in decreasing order). Interestingly, these two additional constraints are affine. The above optimization problem can further be simplified using the following lemma,

**Lemma** **2.**
*The objective function of the optimization problem (37) is optimized when the constraints (37a) and (37c) are satisfied with equality.*


**Proof.** Intuitively, as the objective function is an increasing and decreasing function of each element of $\gamma_{s}$ and $\sigma_{r}^{2}$, respectively, at convergence, the constraints are met with equality. See Appendix C for the proof.    □

### 3.3. Algorithm Description

In this subsection, our proposed algorithm is given. In short, it works as follows. First, based on the task of the IRS in the system, we compute the effect of IRS on the RSI, source–relay and/or relay–destination channel links. After that, we design the best signal design for the source and relays transmitters with the objective of maximizing the throughput. In the rest of this subsection, the detailed explanation of the algorithm is given. First, we need to solve the optimization problem (37). It can be readily shown that $R_{\mathrm{rd}}^{\mathrm{FD}}$ is a monotonically increasing function of $P_{r}$. Furthermore, one can show that $R_{\mathrm{sr}}^{\mathrm{FD}}$ is an increasing function with respect to $P_{s}$ and a decreasing function with respect to $T^{\prime }$ and $P_{r}$ (See Appendix D). Consequently, the worst-case RSI chooses a strategy to reduce the spectral efficiency, while the relay and the source cope with such strategy for improving the system robustness. That means, on one hand, the RSI hurts the stronger eigendirections of the received signal space more than the weaker ones. However, on the other hand, the source tries to cope with this strategy adaptively by smart eigen selection. This process clearly makes the optimization problem complicated at the source–relay hop. Unlike the source–relay hop, the resource allocation problem at the relay–receiver hop is rather easy. Since at the relay–receiver hop there is only one maximization, we can find the sum capacity simply by using the well-known water-filling algorithm.

Observe that although finding each $R_{\mathrm{sr}}^{\mathrm{FD}}$ and $R_{\mathrm{rd}}^{\mathrm{FD}}$ separately is a convex problem, the problem (37) as a whole is not convex. Therefore in this paper, we find the optimal $R_{\mathrm{sr}}^{\mathrm{FD}}$ by keeping $R_{\mathrm{rd}}^{\mathrm{FD}}$ fixed. Then we use the resulting $R_{\mathrm{sr}}^{\mathrm{FD}}$ to find optimal $R_{\mathrm{rd}}^{\mathrm{FD}}$ and again, using the new resulted $R_{\mathrm{rd}}^{\mathrm{FD}}$ to find optimal $R_{\mathrm{sr}}^{\mathrm{FD}}$. This iterative process repeats until the convergence. Our simulation showed that the algorithm has a very fast convergence and only in rare cases does it take more than 20 iterations for the algorithm to converge. This is mainly due to the fact that inequalities (37d) and (37e) restrict the eigenvalues to vary up to a certain limit, which in turn, makes the whole outputs more stable. Figure 4 depicts a typical histogram of iterations. As it can be seen, only less than 3% of cases did not converge until 50 iterations.

Notice that the optimum values for the transmission power on relay hop may not sum to $P_{r}$. The reason is that $R_{\mathrm{sr}}^{\mathrm{FD}}$ is a monotonically decreasing function of $P_{r}$ and as we are interested in the $\min(R_{\mathrm{sr}}^{\mathrm{FD}},R_{\mathrm{rd}}^{\mathrm{FD}})$, with $R_{\mathrm{sr}}^{\mathrm{FD}}<R_{\mathrm{rd}}^{\mathrm{FD}}$ we will have $\min\left(R_{\mathrm{sr}}^{\mathrm{FD}},R_{\mathrm{rd}}^{\mathrm{FD}}\right)=R_{\mathrm{sr}}^{\mathrm{FD}}$. Therefore, it is in our interest to keep $P_{r}$ as low as possible to increase $R_{\mathrm{sr}}^{\mathrm{FD}}$ as much as possible. Analogously, in the case of $R_{\mathrm{sr}}^{\mathrm{FD}}>R_{\mathrm{rd}}^{\mathrm{FD}}$ we have $\min\left(R_{\mathrm{sr}}^{\mathrm{FD}},R_{\mathrm{rd}}^{\mathrm{FD}}\right)=R_{\mathrm{rd}}^{\mathrm{FD}}$ which can be increased by increasing the total power usage of relay’s transmitter. As a result, the well-known bisection method can be used to find the optimal rate where we have $R_{\mathrm{sr}}^{\mathrm{FD}}=R_{\mathrm{rd}}^{\mathrm{FD}}$, unless the case $R_{\mathrm{sr}}^{\mathrm{FD}}\geq R_{\mathrm{rd}}^{\mathrm{FD}}$ happens even if the maximum allowed power is used at the relay transmitter. In such a case, the relay–destination link becomes the bottleneck.

Now we focus on how to find $R_{\mathrm{sr}}^{\mathrm{FD}}$. In order to find the sum rate for the source–relay hop, we assume that we are already given $\gamma_{r}^{⋆}$ which is the vector of relay input powers that maximizes the sum rate at the relay–destination hop. The next step is to complete the minimization over $\sigma_{r}$ and the maximization over $\gamma_{s}$. One approach to solve this problem is to solve it iteratively. With this method, first one finds the optimal $\gamma_{s}$ by solving the maximization part of (37) under the assumption that the optimal $\sigma_{r}$ is given, and then, having the optimal $\gamma_{s}$, the minimization part of (37) can be solved efficiently. This process goes on until the convergence of $\gamma_{s}$ and/or $\sigma_{r}$. The maximization part is performed using the water-filling method. However, the additional conditions $\forall \;i\leq \min\left(M,K_{r}\right),\sigma_{1_{i}}^{2}\gamma_{s_{\rho \left(i\right)}}\geq \sigma_{1_{i+1}}^{2}\gamma_{s_{\rho (i+1)}}$ should be taken into account. For instance, if the optimal value for $\gamma_{s_{i}}$ turns out to be equal to zero, then we should have $\gamma_{s_{j}}=0$ for all j>i irrespective of their SNR. Figure 5 depicts two different examples of multi-level water-filling algorithms. As it can be seen, first, a regular water-filling algorithm is considered where for each subchannel we have $\frac{\sigma_{1_{i}}^{2}}{1+\gamma_{r_{i}}\sigma_{r_{\rho (i)}}^{2}}$ as its channel gain. After finding the water-level in this way, we need to impose $\gamma_{s_{\rho (i+1)}}\leq \frac{\min_{1\leq i^{\prime }\leq i}\left\{\sigma_{1_{i^{\prime }}}^{2}\gamma_{s_{\rho \left(i^{\prime }\right)}}\right\}}{\sigma_{1_{i+1}}^{2}}$. These additional restrictions act like caps on top of the water and create multilevel water-filling which can be interpreted as a cave. Figure 5a shows the case where these caps do not make any subchannel to have zero power. However, Figure 5b shows the case where subchannel i=13 has to be zero as a result of the cap imposed by the additional constraints (37d). In this case, we have $\gamma_{s_{\rho (13)}}=0$, and as a result $\min_{1\leq i^{\prime }\leq 13}\left\{\sigma_{1_{i^{\prime }}}^{2}\gamma_{s_{\rho \left(i^{\prime }\right)}}\right\}=0$. Thus, this condition forces all other subchannels (i.e., i>13) to get no power. Algorithm 1 provides the detail of multilevel water-filling. For the minimization part, a Lagrangian multiplier is used. We have
(38)$$\begin{array}{cc} L= & \sum_{i=1}^{\min(M,K_{r})}\;\log_{2}\;\left(1+\frac{\sigma_{1_{i}}^{2}\gamma_{s_{\rho \left(i\right)}}}{\sigma_{t}^{2}+T_{1}P_{s}+\gamma_{r_{i}}\sigma_{r_{\rho \left(i\right)}}^{2}}\right)+\lambda \left(\sum_{i=0}^{N}\sigma_{r_{i}}^{2}-T_{r}\right). \end{array}$$

Calculating $\frac{\partial L}{\partial \sigma_{r_{i}}^{2}}=0$ we arrive at
(39)$$\begin{array}{c} \sigma_{r_{i}}^{2}\;=\;{\left[\frac{\sqrt{{\left(\sigma_{1_{i}}^{2}\gamma_{s_{i}}\right)}^{2}\;+\;\frac{4\sigma_{1_{i}}^{2}\gamma_{s_{i}}\gamma r_{i}}{\lambda }}\;-\;\sigma_{1_{i}}^{2}\gamma_{s_{i}}\;-\;2\left(\sigma_{t}^{2}\;+\;T_{1}P_{s}\right)}{2\gamma_{r_{i}}}\right]}^{+}, \end{array}$$
where λ is the water level.

Similarly to the maximization case, there are additional constraints $\gamma_{r_{i}}\sigma_{s_{\rho \left(i\right)}}^{2}\geq \gamma_{r_{i+1}}\sigma_{r_{i+1}}^{2}$ that must be considered during the minimization process. However, it can be shown that if the constraints $\gamma_{r_{i}}\geq \gamma_{r_{i+1}}$ and $\sigma_{1_{i}}^{2}\gamma_{s_{\rho \left(i\right)}}\geq \sigma_{1_{i+1}}^{2}\gamma_{s_{i+1}}$ are met, then the constraint $\gamma_{r_{i}}\sigma_{s_{\rho \left(i\right)}}^{2}\geq \gamma_{r_{i+1}}\sigma_{r_{i+1}}^{2}$ becomes redundant. Please refer to Appendix E for proof. The summary of the algorithm to find the achievable rate can be found in Algorithm 2. Next we deal with the optimization for the cases where IRS is utilized to help either the source–relay or relay–destination channels. In such cases, the optimization part over the covariance matrices remains the same as the abovementioned case. In addition, the optimization of the IRS elements can be performed using eigenvalue decomposition and the algorithm introduced in [8]. Notice that for the case in which IRS is assisting the source–relay link, the term $T_{1}P_{s}$ in (27) should be replaced with $\left(T_{1}+T_{SI}T_{IR}\right)P_{s}$, and for the case where IRS helps the relay–destination link, the term $T_{2}P_{s}$ in (28) should be replaced with $\left(T_{2}+T_{RI}T_{ID}\right)P_{r}$. The pseudo code for these scenarios is given in Algorithm 3.
**Algorithm 1** The optimal $\gamma_{s}$1:Find power allocation $P^{0}$ using water-filling algorithm2:**while**$|P^{\left(q\right)}-P^{\left(q-1\right)}|$ is large **do**3:    Define temp=04:    **for** *i* **do**5:        Calculate ${\mathrm{cap}}_{i}=\min_{1\leq i^{\prime }\leq i-1}\left\{\sigma_{1_{i^{\prime }}}^{2}\gamma_{s_{\rho \left(i^{\prime }\right)}}\right\}/\sigma_{1_{i}}^{2}$6:        **if** $P_{i}>{\mathrm{cap}}_{i}$ **then**7:           $P_{i}={\mathrm{cap}}_{i}$8:           $\mathrm{temp}=\mathrm{temp}+P_{i}-{\mathrm{cap}}_{i}$9:        **end if**10:    **end for**11:    $P=P+\frac{\mathrm{temp}}{\mathrm{number}\;\mathrm{of}\;\mathrm{channels}}$12:**end while**

**Algorithm 2** Robust Transceiver Design for FD scenario, the first case
1:Define $U=P_{r}$, L=0, ${\bar{P}}_{r}^{\left(1\right)}=\frac{P_{r}}{2}$2:**while**|U−L| is large **do**3:    Determine $\gamma_{r}^{}={\left[\tau_{r}^{}-\frac{1}{\sigma_{2}^{2}}\right]}^{+}$, s.t. $∥\gamma_{r}^{}{∥}_{1}={\bar{P}}_{r}^{}$4:    Define $\sigma_{r}^{{\left(0\right)}^{2}}=0$ and $\sigma_{r}^{{\left(1\right)}^{2}}=1$ and q=05:     Set $T^{\prime }={\left(\sqrt{T_{r}}-\sigma_{\min}\left(H_{IR}*H_{RI}^{T}\right)\right)}^{2}$6:    **while** $∥\sigma_{r}^{{\left(q\right)}^{2}}-\sigma_{r}^{{\left(q-1\right)}^{2}}{∥}_{1}$ is large **do**7:        Obtain $\sigma_{r}^{{\left(q\right)}^{2}}$, using Equation (39)8:        Obtain $\gamma_{s}^{\left(q\right)}$, using Algorithm 19:        q=q+110:    **end while**11:    Calculate $R_{\mathrm{sr}}^{}$ and $R_{\mathrm{rd}}^{}$12:    **if** $R_{\mathrm{sr}}^{}>R_{\mathrm{rd}}^{}$ **then**13:        $U={\bar{P}}_{r}$14:    **else if** $R_{\mathrm{sr}}^{}<R_{\mathrm{rd}}^{}$ **then**15:        $L={\bar{P}}_{r}$16:    **end if**17:    ${\bar{P}}_{r}=\frac{U+L}{2}$18:
**end while**



**Algorithm 3** Robust Transceiver Design for FD scenario, the second and third case
1:Define $U=P_{r}$, L=0, ${\bar{P}}_{r}^{\left(1\right)}=\frac{P_{r}}{2}$2:Define Case=1 if the IRS is being used help the source–relay link or Case=0 if IRS is being used to help the relay–destination link3:**while**|U−L| is large **do**4:    Determine $\gamma_{r}^{}={\left[\tau_{r}^{}-\frac{1}{\sigma_{2}^{2}}\right]}^{+}$, s.t. $∥\gamma_{r}^{}{∥}_{1}={\bar{P}}_{r}^{}$5:    Define $\sigma_{r}^{{\left(0\right)}^{2}}=0$ and $\sigma_{r}^{{\left(1\right)}^{2}}=1$ and q=06:    **if** Case=1 **then**7:        ${\bar{H}}_{tot_{2}}=\left({\bar{H}}_{1}+{\bar{H}}_{RI}\Theta {\bar{H}}_{IR}\right)$8:        Find the optimum Θ, using Algorithm 1 in [8]9:    **else if** Case=0 **then**10:        ${\bar{H}}_{tot_{3}}=\left({\bar{H}}_{2}+{\bar{H}}_{RI}\Theta {\bar{H}}_{IR}\right)$11:        Find the optimum Θ, using Algorithm 1 in [8]12:    **end if**13:    Obtain $\gamma_{s}^{\left(q\right)}$, using Algorithm 114:    q=q+115:
**end while**
16:Calculate $R_{\mathrm{sr}}^{}$ and $R_{\mathrm{rd}}^{}$17:
**if**

$R_{\mathrm{sr}}^{}>R_{\mathrm{rd}}^{}$

**then**
18:    $U={\bar{P}}_{r}$19:
**else if**

$R_{\mathrm{sr}}^{}<R_{\mathrm{rd}}^{}$

**then**
20:    $L={\bar{P}}_{r}$21:
**end if**
22:

${\bar{P}}_{r}=\frac{U+L}{2}$




### 3.4. Discussion

In this part, we evaluate the various aspects of our method. First, we examine the complexity of our algorithm and compare it with the state of the art. Algorithms 1 and 2 are the main solutions provided in this paper. Algorithm 1 is a multi-level water-filling, and as a result, it has the complexity of $O(I_{w}\min\left(N_{t},K_{r}\right))$, where $I_{w}$ is a constant that is independent of system parameters and is only related to the accuracy of the multi-level water-filling algorithm. Algorithm 2 requires the SVD for matrices $H_{1}$, $H_{2}$ and $\left(H_{IR}*H_{RI}^{T}\right)$, with the complexity $O(N_{t}K_{r}\min\left(N_{t},K_{r}\right))$, $O(N_{r}K_{t}\min\left(N_{r},K_{t}\right))$ and $O(MK_{r}K_{t}\min\left(M,K_{r}K_{t}\right))$, respectively. Furthermore, the Khatri–Rao multiplication $H_{IR}*H_{RI}^{T}$ is needed that has the complexity $O(N_{r}K_{t}M)$. As a result, the overall complexity of our method is $O(MK_{r}K_{t}$$\min\left(M,K_{r}K_{t}\right)+N_{r}K_{t}M+N_{t}K_{r}\min\left(N_{t},K_{r}\right)+N_{r}K_{t}\min\left(N_{r},K_{t}\right)+I_{t}\left(I_{w}\min\left(N_{t},K_{r}\right)\right))$, where $I_{t}$ is a constant independent of the system parameters. Interestingly, our method has a super linear complexity with respect to the number of IRS elements which is better than the state of the art works, e.g., [5,8]. This means that our algorithm in more energy efficient and suitable for latency sensitive applications. It should also be noted that our algorithm does not provide the optimal IRS pattern; instead, it provides analytical bounds for the performance of the IRS that can be used as a benchmark. In other words, our work provides a tool with which one can evaluate the efficiency of their robust design. A comparison between our method and previous works is summarized in Table 2

## 4. Achievable Rate (Half-Duplex Relay)

We consider a simple HD relay where the source and the relay transmit in two subsequent time instances. Notice that for the case of HD, IRS can be used to assist both the source–relay and the relay–destination channels as the signal is being sent over each of these channels in a different time slot. Therefore, the received signals at the relay and the destination can, respectively, be expressed as
(40)$$\begin{array}{cc} y_{r} & =\;\left({\hat{H}}_{1}\;+\;{\hat{H}}_{IR}\Theta {\hat{H}}_{\mathrm{SI}}\right)x_{r}\;+\;\left({\bar{H}}_{1}\;+\;{\bar{H}}_{IR}\Theta {\bar{H}}_{\mathrm{SI}}\right)x_{r}+n_{t}, \end{array}$$
(41)$$\begin{array}{cc} y_{d} & =\;\left({\hat{H}}_{2}\;+\;{\hat{H}}_{ID}\Theta {\hat{H}}_{RI}\right)x_{r}\;+\;\left({\bar{H}}_{2}\;+\;{\bar{H}}_{ID}\Theta {\bar{H}}_{RI}\right)x_{r}+n_{d}. \end{array}$$

Consequently, the achievable rates for the transmitter–relay and relay–destination links can be expressed as below
(42)$$\begin{array}{c} R_{\mathrm{sr}}^{\mathrm{HD}}\;=\;\log|I_{K_{r}}\;+\;\left({\hat{H}}_{1}\;+\;{\hat{H}}_{IR}\Theta {\hat{H}}_{\mathrm{SI}}\right)Q_{s}{\left({\hat{H}}_{1}\;+\;{\hat{H}}_{IR}\Theta {\hat{H}}_{\mathrm{SI}}\right)}^{H}\\ {\left(\sigma_{t}^{2}I_{K_{r}}\;\;+\;\left({\bar{H}}_{1}\;+\;{\bar{H}}_{IR}\Theta {\bar{H}}_{\mathrm{SI}}\right)Q_{s}\left({\bar{H}}_{1}\;+\;{\bar{H}}_{IR}\Theta {\bar{H}}_{\mathrm{SI}}\right){\;}^{H}\;\right)}^{-1}\;|\\ \geq \sum_{i=1}^{\min\left(M,K_{r}\right)}\;\log_{2}\;\left(\;1+\;\frac{\lambda_{i}\left(H_{1}^{\prime }Q_{s}{H_{1}^{\prime }}^{H}\right)}{\;\sigma_{t}^{2}+T_{1}P_{s}}\right),\\ R_{\mathrm{rd}}^{\mathrm{HD}}\;=\;\log|I_{N}\;+\;\left({\hat{H}}_{2}\;+\;{\hat{H}}_{ID}\Theta {\hat{H}}_{\mathrm{RI}}\right)Q_{r}{\left({\hat{H}}_{2}\;+\;{\hat{H}}_{ID}\Theta {\hat{H}}_{\mathrm{RI}}\right)}^{H}\\ {\left(\sigma_{d}^{2}I_{N}\;\;+\;\left({\bar{H}}_{2}\;+\;{\bar{H}}_{ID}\Theta {\bar{H}}_{\mathrm{RI}}\right)Q_{r}{\left({\bar{H}}_{2}\;+\;{\bar{H}}_{ID}\Theta {\bar{H}}_{\mathrm{RI}}\right)}^{H}\right)}^{-1}| \end{array}$$
(43)$$\begin{array}{c} \geq \sum_{i=1}^{\min\left(M,K_{r}\right)}\;\log_{2}\;\left(\;1+\;\frac{\lambda_{i}\left(H_{2}^{\prime }Q_{r}{H_{2}^{\prime }}^{H}\right)}{\;\sigma_{d}^{2}+T_{2}P_{r}}\right). \end{array}$$
where $H_{1}^{\prime }={\hat{H}}_{1}\;+\;{\hat{H}}_{IR}\Theta {\hat{H}}_{\mathrm{SI}}$ and $H_{2}^{\prime }={\bar{H}}_{2}\;+\;{\bar{H}}_{ID}\Theta {\bar{H}}_{\mathrm{RI}}$. In addition, $R_{\mathrm{sr}}^{\mathrm{HD}}$ and $R_{\mathrm{rd}}^{\mathrm{HD}}$ are the achievable rates on the source–relay and relay–destination links, respectively. Using time sharing, the achievable rate between the source and destination nodes is given by
(44)$$\begin{array}{c} R^{\mathrm{HD}}=\min\left(\alpha R_{\mathrm{sr}}^{\mathrm{HD}},\left(1-\alpha \right)R_{\mathrm{rd}}^{\mathrm{HD}}\right), \end{array}$$
where α is the time-sharing parameter.

Note that in half-duplex relaying, the source and relay transmissions are conducted in separate channel uses. Hence, the transmit covariance matrices $Q_{s}\in H^{N_{t}\times N_{t}}$ and $Q_{r}\in H^{K_{t}\times K_{t}}$ are optimized by maximizing the achievable rate from the source to the destination. Here, the convex cone of Hermitian positive semidefinite matrices of dimensions $N_{t}\times N_{t}$ and $K_{t}\times K_{t}$ are represented by $H^{N_{t}\times N_{t}}$ and $H^{K_{t}\times K_{t}}$, respectively. Importantly, for maximizing this achievable rate, the time-sharing parameter, i.e., α, needs to be optimized alongside the system parameters, e.g., power allocation. Readily, optimal α occurs at $\alpha R_{\mathrm{sr}}^{\mathrm{HD}}=\left(1-\alpha \right)R_{\mathrm{rd}}^{\mathrm{HD}}$. Therefore, the achievable rate becomes as follows
(45)$$\begin{array}{c} R^{\mathrm{HD}}=\frac{R_{\mathrm{sr}}^{\mathrm{HD}}R_{\mathrm{rd}}^{\mathrm{HD}}}{R_{\mathrm{sr}}^{\mathrm{HD}}+R_{\mathrm{rd}}^{\mathrm{HD}}}. \end{array}$$

Notice that as the objective function of the above optimization problem is a monotonically increasing function of both $R_{\mathrm{sr}}^{\mathrm{HD}}$ and $R_{\mathrm{rd}}^{\mathrm{HD}}$, the problem can be simplified to maximizing $R_{\mathrm{sr}}^{\mathrm{HD}}$ and $R_{\mathrm{rd}}^{\mathrm{HD}}$ separately.

Next, we provide the solution to the rate optimization problem when IRS is assisting the source–relay link. We have
(46)$$\begin{array}{cc} \max_{Q_{s}}\; & \min_{{\bar{H}}_{SI},{\bar{H}}_{1},{\bar{H}}_{\mathrm{IR}}}\;R_{\mathrm{sr}}^{\mathrm{HD}} \end{array}$$
(46a)$$\begin{array}{cc} \mathrm{subject}\;\mathrm{to}\;\; & \mathrm{Tr}\left(Q_{s}\right)\leq P_{s}, \end{array}$$
(46b)$$\begin{array}{cc} \; & \mathrm{Tr}\left({\bar{H}}_{1}{\bar{H}}_{1}^{H}\right)\leq T_{1}, \end{array}$$
(46c)$$\begin{array}{cc} \; & \mathrm{Tr}\left({\bar{H}}_{SI}{\bar{H}}_{SI}^{H}\right)\leq T_{SI}, \end{array}$$
(46d)$$\begin{array}{cc} \; & \mathrm{Tr}\left({\bar{H}}_{IR}{\bar{H}}_{IR}^{H}\right)\leq T_{IR}, \end{array}$$

The above optimization problem follows the same approach applied for the optimization of the relay–destination link in the FD scenario. As a result, the same method could be applied to find it. In other words, the well-known water-filling algorithm can be used to find the optimal covariance matrices along with the algorithm introduced in [8] to find the best IRS pattern. This process continues iteratively until it finally converges. The solution to $R_{rd}^{HD}$ is the same as well, and the same procedure can be applied to find $Q_{r}$. The overall procedure of finding the solution for the HD mode is summarized in Algorithm 4.
**Algorithm 4** Robust Transceiver Design for HD scenario1:${\bar{H}}_{tot_{2}}=\left({\bar{H}}_{1}+{\bar{H}}_{RI}\Theta {\bar{H}}_{IR}\right)$2:Find the optimum Θ, using Algorithm 1 in [8]3:${\bar{H}}_{tot_{3}}=\left({\bar{H}}_{2}+{\bar{H}}_{RI}\Theta {\bar{H}}_{IR}\right)$4:Find the optimum Θ, using Algorithm 1 in [8]5:Find $R_{\mathrm{sr}}^{\mathrm{HD}}$, using Equation (42)6:Find $R_{\mathrm{rd}}^{\mathrm{HD}}$, using Equation (43)7:Find $R^{\mathrm{HD}}$, using Equation (45)

## 5. Numerical Results

We assume the transmit power budgets at the source and at the relay are $P_{s}=5$ and $P_{r}=1$, respectively. Moreover, the AWGN spectral density is assumed to be −175 dBm and the bandwidth is BW=180 MHz. In this section, we investigate the performance of IRS-assisted full-duplex relaying with RSI channel uncertainty bound $T_{r}$, i.e., $\mathrm{Tr}\left({\bar{H}}_{r}{\bar{H}}_{r}^{H}\right)\leq T_{r}$. We consider all the channels to follow the Rician distribution with the factor ϵ=0.1 and the specificaiton given in Table 3. We also assume $T_{x}=0.001,x\in \left\{1,2,SI,IR,RI,ID\right\}$. We perform Monte Carlo simulations with $L=10^{3}$ realizations from random channels and noise vectors. Hence, the average worst-case throughput rate is defined as the average of worst-case rates for *L* randomization, i.e., $R_{\mathrm{av}}=\frac{1}{L}\sum_{l=1}^{L}R_{l}.$ Notice that for each set of realizations, we solve the robust transceiver design as is elaborated in Algorithm 2. We run different sets of simulations as described in the following subsections.

### 5.1. Antenna Array Increment with No IRS

In this part, first we assume that there is no IRS installed. Then we evaluate the performance of the system using different strategies. Thereafter, we examine how installing an IRS can help increase the throughput. We consider two cases where the source, relay and destination are equipped with (a) a small antenna array, and (b) a large antenna arrays. In order to see the impact of IRS, we first assume that there is no IRS installed. For these cases, we have

(a)$N_{t}=4,K_{r}+K_{t}=10,N_{r}=4$,(b)$N_{t}=10,K_{r}+K_{t}=24,N_{r}=10$.

These cases are considered to highlight the performance of full-duplex DF relaying as a function of the number of antennas with the worst-case RSI. Interestingly, as the number of antennas at the source, relay and destination increase, full-duplex relaying achieves a higher throughput rate even with strong RSI. This can be seen by comparing rates from Figure 6a to those from Figure 6b.

Furthermore, notice that the worst-case RSI casts strong interference on the strong streams from the source to the destination. With very low RSI power $T_{r}\to 0$, full-duplex almost doubles the throughput rate compared to the half-duplex counterpart. This can be seen in Figure 6, where the curves have their intercept point with the vertical axis. However, as $T_{r}$ increases, the efficiency of full-duplex operation drops. It is worth noting that at low RSI power the DoF plays the most important role to have a higher sum rate. For instance, consider Figure 6a in which the cases FD={4,5,5,4}, FD={4,4,6,4} and FD={4,6,4,4} have ${\mathrm{DoF}}_{\mathrm{total}}=4$ – ${\mathrm{DoF}}_{\mathrm{total}}$ is the minimum of the DoF of source–relay and relay–destination channels, i.e., ${\mathrm{DoF}}_{\mathrm{total}}=\min\left({\mathrm{DoF}}_{\mathrm{sr}},{\mathrm{DoF}}_{\mathrm{rd}}\right)$ – while the cases FD={4,7,3,4} and FD={4,3,7,4} have ${\mathrm{DoF}}_{\mathrm{total}}=3$. At $T_{r}=0$, there is a noticeable gap between the first three cases and the last two, while the difference of the first three cases from each other is small. The big gap is due to the difference in ${\mathrm{DoF}}_{\mathrm{total}}$, and the small one is due to the difference in SNR. Similarly, in Figure 6b, the three cases FD={10,12,12,10}, FD={10,10,14,10} and FD={10,10,14,10} with ${\mathrm{DoF}}_{\mathrm{total}}=10$ have higher rates than the two cases FD={10,6,18,10} and FD={10,18,6,10} with ${\mathrm{DoF}}_{\mathrm{total}}=6$.

Finally, it can be seen in both Figure 6a,b that at $T_{r}=0$ there is no difference between cases that have the same ${\mathrm{DoF}}_{\mathrm{total}}$ but different ${\mathrm{DoF}}_{\mathrm{sr}}$ and ${\mathrm{DoF}}_{\mathrm{rd}}$. As it can be seen in both Figure 6a,b, for cases with $K_{t}>K_{r}$ the sum rate drops quickly as RSI increases. In fact, the more relative antennas at the relay transmitter compared to its receiver, the faster the sum rate drops with the rise in RSI. To understand this behaviour of the system better, again, consider case {10,18,6,10} and also suppose $T_{r}\to \infty$. As discussed before, we have ${\mathrm{DoF}}_{\mathrm{sr}}=10$ and ${\mathrm{DoF}}_{\mathrm{rd}}=6$. Moreover, we have ${\mathrm{DoF}}_{I}=6$ for the interference channel (${\bar{H}}_{r}$). Unlike the case with no interference, in this case the bottleneck is no longer the relay–destination link. This is due to the fact that interference can act to the detriment of some six of the source–relay subchannels. As we have ${\mathrm{DoF}}_{I}=6$, interference can choose at most six independent subchannels, and as we assumed $T_{r}\to \infty$, for those subchannels we obtain SINR→0. Therefore, no information can be conveyed from those links, and the bottleneck becomes the source–relay link with 4 usable subchannels. It can be seen in Figure 6a,b that as $T_{r}$ increases, the cases with the same sum-rate at $T_{r}=0$ start to diverge because of the different characteristics of the interference they experience. We explain the effect of interference in the following subsection in more detail.

### 5.2. The Impact of IRS

In this part, we evaluate the impact of IRS on the throughput rate when it is used to perform different tasks. Figure 7 shows the throughput for three different scenarios, namely, when IRS is used to help the transmitter–relay link, when it is applied to cancel RSI and when the IRS job is to help the relay–destination link. Then the results are compared with two cases where the system is working in HD with IRS and the case where the system is working in FD with no IRS. It is also assumed that $\frac{T_{r}}{\left(H_{r}H_{r}^{H}\right)}=75\%$, i.e., the system works at high RSI range. As it can be seen, the highest performance is achieved when the IRS is utilized to deal with the RSI. As a result, for the rest of the paper we use the IRS for this purpose. 

Figure 8 shows the impact of IRS on the throughput. For Figure 8a, we considered the case $\left\{N_{t},K_{t},K_{r},N_{r}\right\}=\left\{4,5,5,4\right\}$, and for Figure 8b, we considered $\left\{N_{t},K_{t},K_{r},N_{r}\right\}=\left\{10,12,12,10\right\}$. As shown in the figure, the number of IRS elements has a great impact on RSI cancellation to the extend that having an IRS with M=100 and M=300 can cancel interference of $\frac{T_{r}}{\mathrm{Tr}(H_{r}H_{r}^{H})}=0.75$ for $\left\{N_{t},K_{t},K_{r},N_{r}\right\}=\left\{4,5,5,4\right\}$, and $\left\{N_{t},K_{t},K_{r},N_{r}\right\}=\left\{10,12,12,10\right\}$, respectively. Further, it can also be seen in the figure that having IRS with 20 and 100 elements for the small and large antenna array cases, respectively, is not helpful at all. This is mainly due to the fact that unlike the average case, for the case of worst-case scenario, the number of IRS elements should be at least as large as the dimension of ${\bar{H}}_{r}$. Otherwise, the IRS feasible set cannot span into all dimensions of ${\bar{H}}_{r}$. Therefore, there is always at least one representation for ${\bar{H}}_{r}$ in which IRS cannot perform any RSI cancellation. In addition, comparing two figures Figure 8a,b one can conclude that, when the dimension of ${\bar{H}}_{r}$ increases, the effort that IRS has to make in order to cancel RSI remarkably increases which is consistent with the previous statement.

### 5.3. Relay Tx/Rx Antenna allocation

Suppose that the relay has $K_{t}+K_{r}=8$ in total. Furthermore, following cases in which the number of antenna at the source and destination are $\left\{N_{t},N_{r}\right\}=\left\{4,4\right\}$. The question is, from eight antennas at the relay, how many should be used for reception for the robust design? Figure 9 shows the sum rate as a function of $K_{r}$ for different values of *T* where there is no IRS and there is an IRS with M=60 elements, respectively. As it can be seen, by using more antennas for reception than for transmission, i.e., $K_{r}>K_{t}$, at the relay, the throughput rate is maximized. This is due to the fact that increasing the signal-to-noise ratio (SNR) of the source–relay streams enhances the overall throughput rate more than increasing the number of antennas for transmission in order to enhance the DoF of the relay–destination link. Furthermore, notice that in this setup the overall DoF from the source to destination is limited by the DoF of the source–relay link, i.e., the bottleneck is in the first hop.

By comparing two scenarios, we see that having an IRS not only improves the rates in all cases, but also it may change the best antenna allocation. For instance, for the case of T=15%, it is best to have six antennas at the relay receiver and four4 antennas at the relay transmitter. However, after establishing the IRS, the best antenna allocation changes to five antennas at each end. For instance, the results show that although the ${\mathrm{DoF}}_{\mathrm{total}}$ for both $\left\{N_{t},K_{t},K_{r},N_{r}\right\}=\left\{4,3,5,4\right\}$ and $\left\{N_{t},K_{t},K_{r},N_{r}\right\}=\left\{4,5,3,4\right\}$ is three, the sum rate capacity of the latter is much better than that of the former at high interference. This is because of the fact when ${\mathrm{DoF}}_{\mathrm{sr}}>{\mathrm{DoF}}_{\mathrm{rd}}$, the source–relay link enjoys ${\mathrm{DoF}}_{\mathrm{sr}}-{\mathrm{DoF}}_{\mathrm{rd}}$ subchannels with no interference. Therefore, the source can manage to obtain a higher sum rate by choosing its power allocation wisely. However, in the case of ${\mathrm{DoF}}_{\mathrm{sr}}\leq {\mathrm{DoF}}_{\mathrm{rd}}$, no matter how well the power allocation is performed, all sub channels suffer from interference at the source–relay end.

### 5.4. Full-Duplex vs. Half-Duplex

In this subsection, we determine the thresholds where the HD relaying outperforms the FD relaying. This threshold provides a mode-switching threshold in hybrid HD/FD relay systems. As it can be seen in Figure 10, for each case of $K_{r}$, there is a maximum value of $\frac{T}{P}$ above which the HD mode outperforms the FD mode in terms of sum rate maximization. Furthermore, Figure 10 shows the threshold for different IRS configurations. For this part, we continued with the case of $N_{t}=4,K_{r}=5,K_{t}=5,N_{r}=4$. As it can be seen, by increasing the number of antennas, the threshold occurs at higher RSI. This is in fact a direct result of obtaining better performance by having more antennas at the relay’s receiver. It is worth noting that the IRS has a great impact on the performance of FD relaying. For instance, by having an IRS consisting of only 60 elements, the FD mode outperforms the HD mode in almost all cases.

## 6. Conclusions

In this paper, we investigated a multi-antenna source communicating with a multi-antenna destination through a multi-antenna relay. The relay is assumed to exploit a decode-and-forward (DF) strategy. An IRS is installed to help the relay cope with the RSI. The transceivers are designed in order to be robust against the worst-case residual self-interference (RSI). To this end, the worst-case achievable throughput rate is maximized. This optimization problem turns out to be a non-convex problem. Assuming that the degrees-of-freedom (DoF) of the source–relay link is less than the DoF of the relay–destination link, we determined the left and right matrices of the singular vectors of the worst-case RSI channel. Then, the problem is simplified to the optimal power allocation at the transmitters, which guarantees robustness against the worst-case RSI singular values. This simplified problem is still non-convex. Based on the intuitions for optimal power allocation at the source and relay, we proposed an efficient algorithm to capture a stationary point. Our proposed method showed a significant improvement in robustness. More precisely, we showed that in the case of high uncertainty, using our method can lead to at least 100% worst-case throughput improvement for the case of few antenna arrays and up to 500% for the case of large antenna arrays at transceivers. Furthermore, we confirmed that there is a direct relation between the performance of the system and the number of IRS elements. The simulations show that having the IRS with as low as 90 and 300 elements can completely remove the RSI for our system configuration. Finally, we showed that when there is no RSI, the impact of the relay can be fully harnessed where the number of antennas are equal at the relay transmitter and receiver. Therefore, employing the IRS to deal with the RSI can lead to the best performance of the relay.

## Figures and Tables

**Figure 1 sensors-22-06587-f001:**
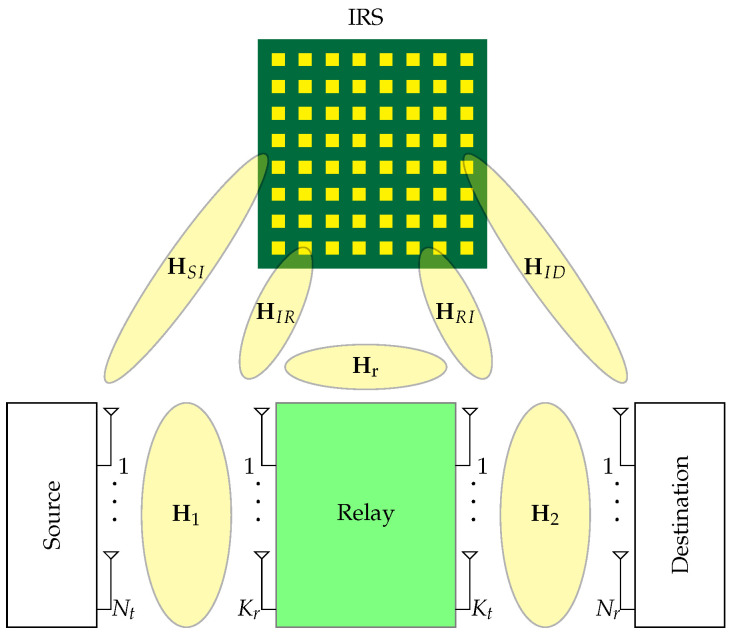
System model of an IRS assisted full-duplex relay. In our model both source and destination are equipped with $N_{t}$ and $N_{r}$ antennas, respectively. In addition, the relay is equipped with $K_{t}$ transmitting and $K_{t}$ receiving antennas, and the IRS has *M* passive elements.

**Figure 2 sensors-22-06587-f002:**
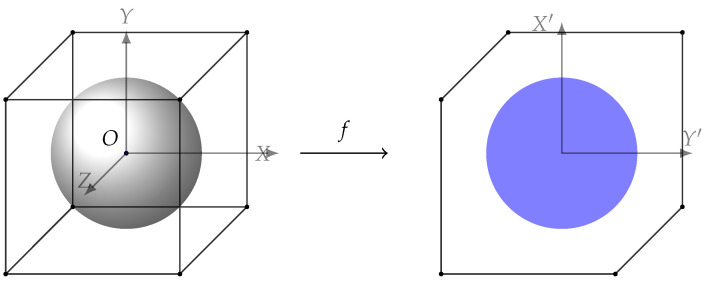
An example of mapping from three-dimensional to two-dimensional space. The hexagon and the circle are the mapping output of the cube and the sphere respectively.

**Figure 3 sensors-22-06587-f003:**
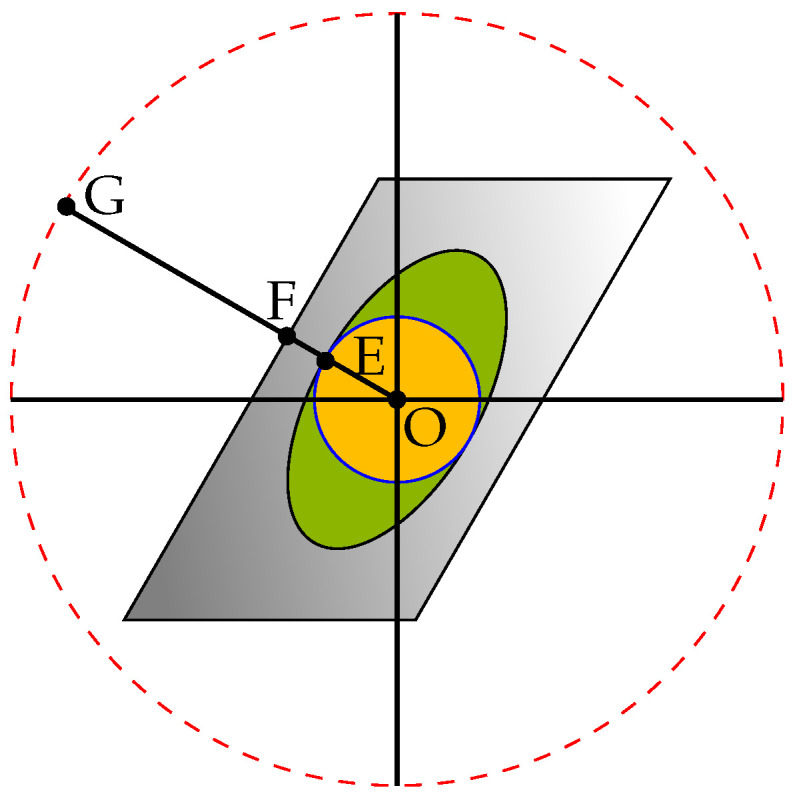
Geometrical representation of optimization problem (18).

**Figure 4 sensors-22-06587-f004:**
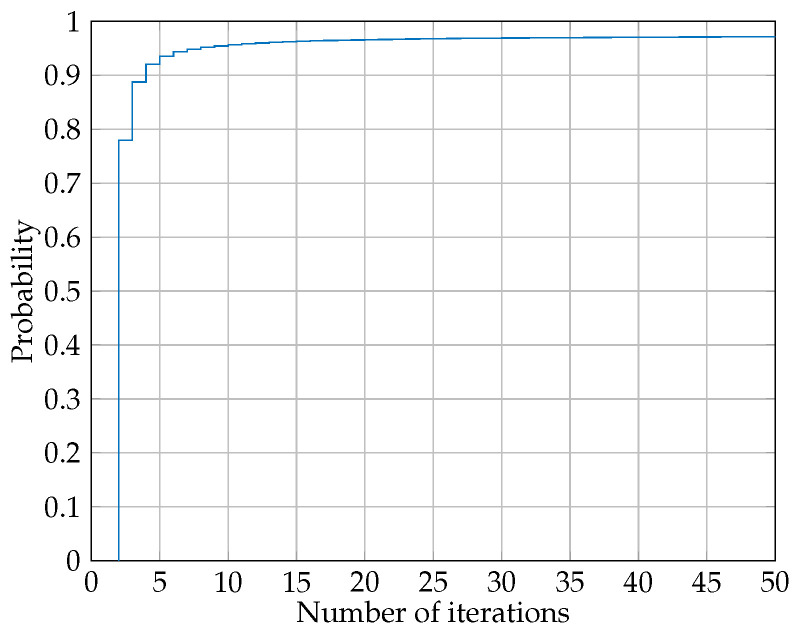
Cumulative distribution function (cdf) of iterations when $P_{s}=5$ and $P_{r}=1$ and $M=K_{t}=K_{r}=N=10$. The maximum number of iterations is set to be 50. Cases that took more than 50 iterations to converge are considered to be divergent.

**Figure 5 sensors-22-06587-f005:**
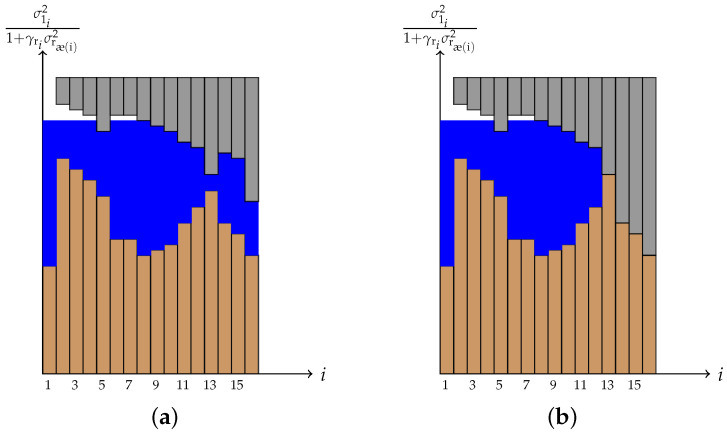
Examples of multilevel water-filling for two different cases. (**a**) No subchannel with optimum power equal to zero. Note that due to the power cap constraint (37d), water does not have the same level for all subchannels. (**b**) Subchannel i=13 receives the optimum of zero as its input power. Notice that in this case, due to the additional power cap constraint (37d), all the remaining subchannels i>13 also have zero power.

**Figure 6 sensors-22-06587-f006:**
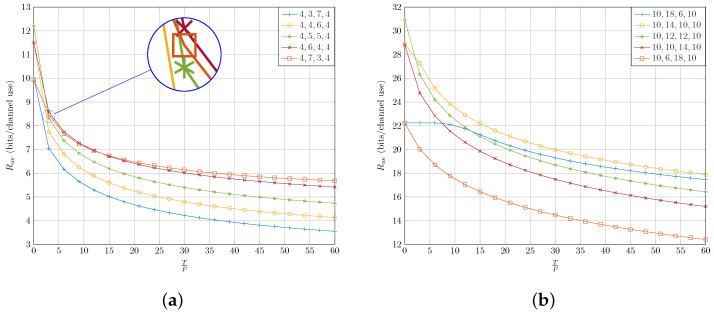
Sum rate throughput of the FD mode as a function of normalized interference $\frac{T}{P}$ with different number of antennas at the source, relay and destination. The transmit power budget at the source and the relay are assumed to be equal, i.e., $P_{s}=5$ and $P_{r}=1$. (**a**) $\left\{N_{t},K_{r}+K_{t},N_{r}\right\}=\left\{4,10,4\right\}$. (**b**) $\left\{N_{t},K_{r}+K_{t},N_{r}\right\}=\left\{10,24,10\right\}$.

**Figure 7 sensors-22-06587-f007:**
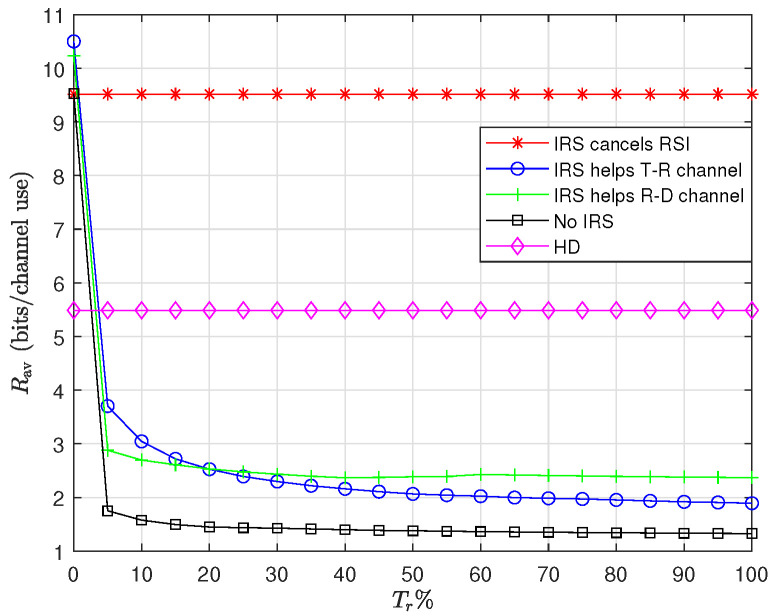
Comparison of five different scenarios: HD with IRS, FD with no IRS, FD with IRS as RSI cancelator; FD with IRS to help transmitter–relay link; FD with IRS to help relay–destination link. We considered the case where $\left\{N_{t},K_{t},K_{r},N_{r},M\right\}=\left\{4,5,5,4,100\right\}$.

**Figure 8 sensors-22-06587-f008:**
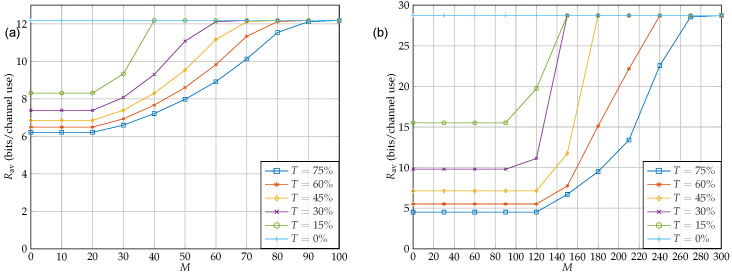
Sum rate throughput as a function of IRS elements. The transmit power budget at the source and the relay are assumed to be equal, i.e., $P_{s}=5$ and $P_{r}=1$: (**a**) {$N_{t}$, $K_{t}$, $K_{r}$, $N_{r}$} = {4, 5, 5, 4}; (**b**) {$N_{t}$, $K_{t}$, $K_{r}$, $N_{r}$} = {10, 12, 12, 10}.

**Figure 9 sensors-22-06587-f009:**
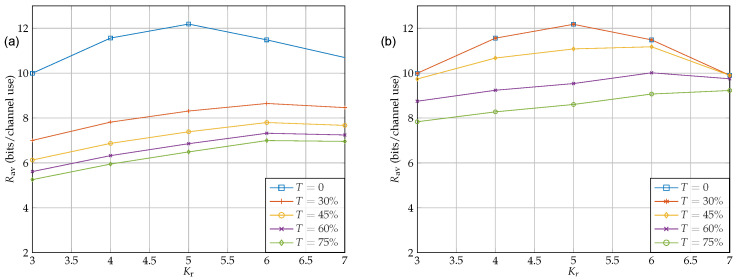
Sum rate throughput as a function of relay receiver antennas $K_{r}$ with and without RSI. The transmit power budget at the source and the relay are assumed to be equal, i.e., $P_{s}=5$ and $P_{r}=1$: (**a**) {$N_{t}$, $K_{t}$ + $K_{r}$, $N_{r}$} = {4, 10, 4} with no IRS; (**b**) {$N_{t}$, $K_{t}$ + $K_{r}$$N_{r}$} = {4, 10, 4} with M = 60.

**Figure 10 sensors-22-06587-f010:**
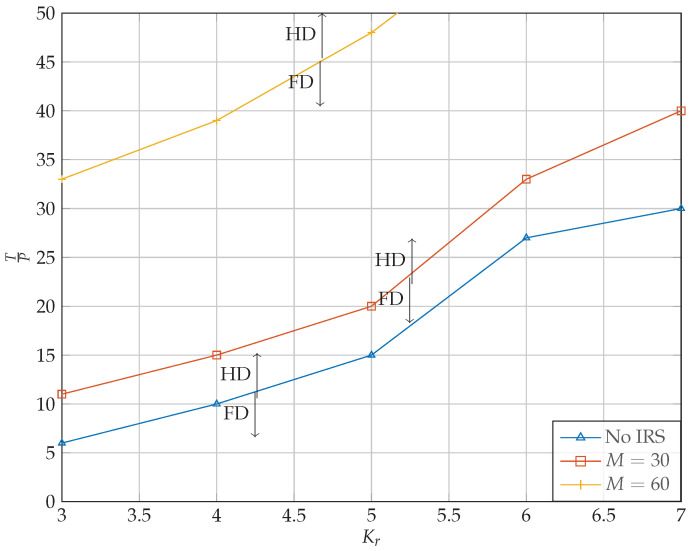
Thresholds for different $K_{r}$ and *M*. The region above each curve indicates values of $\frac{T}{P}$ for which HD outperforms FD. In contrast, points below the curve belong to cases where FD performs better than HD.

**Table 1 sensors-22-06587-t001:** Simulation parameters.

Notation	Definition	Notation	Definition
$N_{t}$	Number of antennas at the source	Θ	IRS phase profile
$N_{r}$	Number of antennas at the destination	$x_{s}$, $x_{r}$	Source and relay transmit signals
$K_{t}$	Number of antennas at the relay’s transmitter	$n_{r}$, $n_{d}$	Additive noise at the relay and destination
$K_{r}$	Number of antennas at the relay’s receiver	$y_{r}$, $y_{d}$	Received signals at the relay and destination, respectively
*M*	Number of IRS elements	κ	FD/HD mode indicator
$H_{r}$, ${\hat{H}}_{r}$, ${\bar{H}}_{r}$	Self-interference actual channel, estimated channel and channel error, respectively	$R_{\mathrm{sr}}^{\mathrm{FD}}$, $R_{\mathrm{rd}}^{\mathrm{FD}}$, $R^{\mathrm{FD}}$	Source–relay, relay–destination and the overall throughput in FD mode
$H_{1}$, ${\hat{H}}_{1}$, ${\bar{H}}_{1}$	Source–Relay actual channel, estimated channel and channel error, respectively	$R_{\mathrm{sr}}^{\mathrm{HD}}$, $R_{\mathrm{rd}}^{\mathrm{HD}}$, $R^{\mathrm{HD}}$	Source–relay, relay–destination and the overall throughput in HD mode
$H_{2}$, ${\hat{H}}_{2}$, ${\bar{H}}_{2}$	Relay–Destination actual channel, estimated channel and channel error, respectively	$Q_{s}$, $Q_{r}$	Source and relay covarriance matrices
$H_{SI}$, ${\hat{H}}_{SI}$, ${\bar{H}}_{SI}$	Source–IRS actual channel, estimated channel and channel error, respectively	$H_{tot_{1}}$	$H_{tot_{1}}=\left({\bar{H}}_{r}+H_{RI}\Theta H_{IR}\right)$
$H_{IR}$, ${\hat{H}}_{IR}$, ${\bar{H}}_{IR}$	IRS–Destination actual channel, estimated channel and channel error, respectively	${\hat{H}}_{tot_{2}}$, ${\bar{H}}_{tot_{2}}$	${\hat{H}}_{tot_{2}}=\left({\hat{H}}_{1}+H_{RI}\Theta {\hat{H}}_{IR}\right)$ ${\bar{H}}_{tot_{2}}=\left({\bar{H}}_{1}+{\bar{H}}_{RI}\Theta {\bar{H}}_{IR}\right)$
$H_{RI}$, ${\hat{H}}_{RI}$, ${\bar{H}}_{RI}$	Relay–IRS actual channel, estimated channel and channel error, respectively	${\hat{H}}_{tot_{3}}$, ${\bar{H}}_{tot_{3}}$	${\hat{H}}_{tot_{3}}=\left({\hat{H}}_{2}+H_{RI}\Theta {\hat{H}}_{IR}\right)$ ${\bar{H}}_{tot_{3}}=\left({\bar{H}}_{2}+{\bar{H}}_{RI}\Theta {\bar{H}}_{IR}\right)$
$H_{IR}$, ${\hat{H}}_{IR}$, ${\bar{H}}_{IR}$	IRS–Relay actual channel, estimated channel and channel error, respectively	$\lambda_{i}\left(X\right)$	*i*’th largest eigenvalue of matrix *X*
$\gamma_{s_{i}}$	$\gamma_{s_{i}}=\lambda_{i}\left(Q_{s}\right)$	$\sigma_{1_{i}}^{2}$	$\sigma_{1_{i}}^{2}=\lambda_{i}\left(H_{1}H_{1}^{H}\right)$
$\gamma_{r_{i}}$	$\gamma_{r_{i}}=\lambda_{i}\left(Q_{r}\right)$	$\sigma_{r_{i}}^{2}$	$\sigma_{r_{i}}^{2}=\lambda_{i}\left({\bar{H}}_{tot_{1}}{\bar{H}}_{tot_{1}}^{H}\right)$
$T_{r}$	The RSI channel uncertainty bound	$\sigma_{2_{i}}^{2}$	$\sigma_{2_{i}}^{2}=\lambda_{i}\left(H_{2}H_{2}^{H}\right)$
$T_{x}$	Channel estimation error bound $x\in \left\{1,2,SI,IR,RI,ID\right\}$	$T^{\prime }$	The remaining RSI channel uncertainty after considering the impact of the IRS

**Table 2 sensors-22-06587-t002:** Comparison of the proposed method with previous studies.

Method	Complexity	Robust Design *	Solution	System Model
This work	$O(MK_{r}K_{t}\min\left(M,K_{r}K_{t}\right))$	Yes	Analytical bounds	Relay and IRS
Zhang et al. [8]	$O(I_{O}M^{3})$	No	Practical solution	Only IRS
Esmaeili et al. [34]	$O(I_{t}\left(I_{w}\min\left(N_{t},K_{r}\right)\right))$	Yes	Practical solution	Only relay
Obeed et al. [9]	$O(I_{\lambda }I_{O}I_{w}M^{3})$	No	Practical solution	Relay and IRS

* This means that the method is robust against channel uncertainties and/or RSI.

**Table 3 sensors-22-06587-t003:** Simulation parameters.

Parameters	Values
Transmitter location	(0m,0m)
IRS location	(4000m,20m)
Relay location	(4000m,0m)
Receiver location	(8000m,0m)
Path-loss	32.6+36.7log(d)
Transmission bandwidth *B*	180Mb

## Data Availability

The data can be provided by the authors H.E. upon reasonable request.

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
