# Peer review of "Robust Transceiver Design for IRS-Assisted Cascaded MIMO Communication Systems"

_sensors, 2022, doi:10.3390/s22176587_

Round 1
Reviewer 1 Report
This paper proposes a transceiver design of a MIMO system and used IRS in two scenarios. 1- IRS is used to eliminate the residual self-interference 2- IRS is used to enhance the quality of the channel. The system model for both scenarios is presented. The channel is optimized for achievable rate in half and full-duplex modes. The paper is well-organized and the results are promising. Also, IRS-assisted communication is a hot topic in wireless networks. So, I would recommend the paper for publication. Nevertheless, I have a few minor comments to help the authors improve the paper.
1- Although some abbreviations like IRS and DF are defined in the abstract RSI is not defined.
2- I would move this sentence from the abstract to the introduction section:
To the best of our knowledge, this is the first work that uses the IRS for RSI cancellation in MIMO full-duplex DF relay systems.
3- The first paragraph of the System mode section is not clear to all readers. Explain clearly why some links have a negligible impact on the throughput (I would suggest electromagnetic reasoning).
4- Figures 3, 4, and 7 are too big to try to make them more compact instead use a bigger font size.
5- The conclusion can be more informative with technical data. Try to summarize the achievements with numbers. For example, throughput, power budget, IRS elements, optimization elements and etc.
Author Response
1- Although some abbreviations like IRS and DF are defined in the abstract RSI is not defined.
We have addressed this issue in the new draft.
2- I would move this sentence from the abstract to the introduction section:
To the best of our knowledge, this is the first work that uses the IRS for RSI cancellation in MIMO full-duplex DF relay systems.
We have addressed this issue in the new draft.
3- The first paragraph of the System mode section is not clear to all readers. Explain clearly why some links have a negligible impact on the throughput (I would suggest electromagnetic reasoning).
This part is rewritten, the changes are in red.
4- Figures 3, 4, and 7 are too big to try to make them more compact instead use a bigger font size.
In the new version, these figures are the same size as the rest.
5- The conclusion can be more informative with technical data. Try to summarize the achievements with numbers. For example, throughput, power budget, IRS elements, optimization elements and etc.
The conclusion is rewritten. The added parts are in red and the removed ones are in blue.

Reviewer 2 Report
Robust Transceiver Design for IRS-Assisted Cascaded MIMO
Systems
Comments
1. In the introduction section, the authors need to write the contribution of the study in the objectives and also need to include the organization of the study.
2. The introduction of the study is very confusing, there are no proper headings regarding which content has been addressed. Moreover, the authors have included the literature of the review in the introduction section. So authors are required to properly categorize the content into the sub-section.
3. The study needs to include more latest relevant research articles to justify the novelty and identified the research gap.
4. In the abstract, the authors are required to rewrite the abstract by focusing on the problem statement, motivation of the study, limitations of the study on the same problem statement, objective of the study, methodology, numerical findings, and novelty.
5. The conclusion section needs to rewrite, where the author needs to properly discuss the study in scientific language with the novelty of the review and future scope.
6. In the study, the authors need to avoid long paragraphs, to avoid the less readability of the article.
7. A comparative table analysis must be included in the article by comparing it with the previous studies.
Author Response
- In the introduction section, the authors need to write the contribution of the study in the objectives and also need to include the organization of the study.
The introduction is rewritten. The added sentences are marked in red and the ones are in blue.
- The introduction of the study is very confusing, there are no proper headings regarding which content has been addressed. Moreover, the authors have included the literature of the review in the introduction section. So authors are required to properly categorize the content into the sub-section.
The introduction is rewritten. The added sentences are marked in red and the ones are in blue.
The study needs to include more latest relevant research articles to justify the novelty and identified the research gap.
New studies and more relevant works has been added to the new draft.
In the abstract, the authors are required to rewrite the abstract by focusing on the problem statement, motivation of the study, limitations of the study on the same problem statement, objective of the study, methodology, numerical findings, and novelty.
The abstract is changed now. The added sentences are marked in red and the ones are in blue.
The conclusion section needs to rewrite, where the author needs to properly discuss the study in scientific language with the novelty of the review and future scope.
The conclusion is rewritten where the results are provided in terms of percentage and numbers.
In the study, the authors need to avoid long paragraphs, to avoid the less readability of the article.
We tried to address this issue in new draft attached.
A comparative table analysis must be included in the article by comparing it with the previous studies.
A table is added in page 16 where the comparison between this work and previous studies is available.

Reviewer 3 Report
This paper investigated a robust transceiver design problem, which maximizes the throughput rate corresponding to the worst-case RSI under a self-interference channel uncertainty bound constraint. The yielded problem turns out to be a non-convex optimization problem, where the non-convex objective is optimized over the cone of semi-definite matrices. Then the authors proposed a multi-level water-filling algorithm to obtain a locally optimal solution iteratively. In light of the evaluation, this paper discussed an attractive topic and with certain innovations. Some of the minor issues are highlighted as follows.
1. The state of the art of this paper needs to be enhanced. In this paper, the yielded problem turns out to be a non-convex optimization problem, which is difficult to obtain the optimal solutions. Since AI becomes a promising solution for solving optimization problems, it is recommended for the authors to review some related works. Just to name a few:
For instance, considering that RIS configurations and resource allocation are prone to be done in near-real-time to meet the constraint in practice, introducing deep learning to address these challenges (such as solving optimization problems efficiently via multi-task learning in [1] and achieving optimal RISs ON-OFF decisions via spectrum learning in [2]) has become a good candidate to address these challenges.
Reconfigurable Intelligent Surface-Assisted Aerial-Terrestrial Communications via Multi-Task Learning, IEEE Journal on Selected Areas in Communications, vol. 39, no. 10, pp. 3035-3050, Oct. 2021.
2. Another concern is the complexity involved. During the implementation, I am curious that how much cost (e.g., delay and energy cost) will be introduced for solving the formulated problems via the proposed iteration-based method, especially for the power-constrained devices cases and latency-sensitive scenarios? This kind of evaluation plays an important role since the aims of the proposed scheme and algorithms address the issues raised in practical scenarios, which makes the research meaningful.
3. The authors may want to present the overall diagram of the proposed algorithm to make the readers easy to follow. Also, the pros and cons of the proposed method need to be highlighted, so other people can not only understand what the original contribution is but also get inspired.
4. There are so many mathematic symbols, which make it very hard to follow. The authors may want to avoid some of them or add a table to make the paper clear. Also, the authors may need to proofread the paper and improve the typesetting, e.g., “We propose a closed-from lower bound” should be “We propose a closed-form lower bound” in the abstract.
Author Response
- The state of the art of this paper needs to be enhanced. In this paper, the yielded problem turns out to be a non-convex optimization problem, which is difficult to obtain the optimal solutions. Since AI becomes a promising solution for solving optimization problems, it is recommended for the authors to review some related works. Just to name a few:
For instance, considering that RIS configurations and resource allocation are prone to be done in near-real-time to meet the constraint in practice, introducing deep learning to address these challenges (such as solving optimization problems efficiently via multi-task learning in [1] and achieving optimal RISs ON-OFF decisions via spectrum learning in [2]) has become a good candidate to address these challenges.
Reconfigurable Intelligent Surface-Assisted Aerial-Terrestrial Communications via Multi-Task Learning, IEEE Journal on Selected Areas in Communications, vol. 39, no. 10, pp. 3035-3050, Oct. 2021.
The introduction is now rewritten. We have included your suggestion along with some other new works in this domain. The new draft is attached
- Another concern is the complexity involved. During the implementation, I am curious that how much cost (e.g., delay and energy cost) will be introduced for solving the formulated problems via the proposed iteration-based method, especially for the power-constrained devices cases and latency-sensitive scenarios? This kind of evaluation plays an important role since the aims of the proposed scheme and algorithms address the issues raised in practical scenarios, which makes the research meaningful.
A whole section related to the complexity of our algorithm along with that of previous works is added that can be found in pages 14 and 16.
- The authors may want to present the overall diagram of the proposed algorithm to make the readers easy to follow. Also, the pros and cons of the proposed method need to be highlighted, so other people can not only understand what the original contribution is but also get inspired.
All algorithms are updated now. We added algorithms belonging to each part seperately. Moreomer a short description of the whole algorithm is added in page 12.
- There are so many mathematic symbols, which make it very hard to follow. The authors may want to avoid some of them or add a table to make the paper clear. Also, the authors may need to proofread the paper and improve the typesetting, e.g., “We propose aclosed-fromlower bound” should be “We propose a closed-form lower bound” in the abstract.
We tried to correct this issues in the new draft. The added words and sentences are marked in red and the removed ones are in blue.
